# Regret Minimization With a Crowd of Awakening Experts

**Anna Lunghi** [* 1]  **Gianmarco Genalti** [* 1]  **Alberto Marchesi** [1]  **Matteo Castiglioni** [1]

## Abstract

We study the *Awakening Crowd of Experts* (ACE) problem, an online learning problem where the set of experts available to the learner grows at each round. ACE is a special case of the well-known sleeping experts problem (Kleinberg et al., 2010), where the number of experts is *huge* ($K = T$). Existing results on sleeping experts preclude any learner from achieving a sublinear regret when the number of available experts is linear in $T$. Inspired by real-world applications, such as Q&A platforms and social proof marketing, we thus focus on the *awakening* version of the sleeping experts problem, where a new expert arrives at every round and never leaves. We show that in the stochastic version of ACE, it is possible to obtain regret $\widetilde{\mathcal{O}}(T^{2/3})$ using an unusual *pessimism* in the face of the uncertainty principle. Moreover, we characterize the dependence of the regret on the stability of an optimal strategy. For both results, we present matching lower bounds. Surprisingly, the adversarial version of ACE is sensibly harder. In particular, we provide a lower bound precluding sublinear $\alpha$-regret for a constant $\alpha$. We provide an algorithm to face this crucial trade-off between competitive ratio and regret, and bound its $\alpha$-regret, almost matching the aforementioned lower bound. As a corollary, we get a $\widetilde{\mathcal{O}}(\log \log T)$ competitive ratio when an optimal strategy enjoys a reward linear in $T$.

## 1. Introduction

The *experts* problem (Littlestone & Warmuth, 1994; Cesa-Bianchi et al., 1997) stands as a cornerstone of online learning, offering a robust framework for decision-making under uncertainty. Despite its relevance, the standard formulation often relies on the idealized assumption that the set of available experts remains static throughout the time horizon. This limitation is partially addressed by the *sleeping experts* framework (Kleinberg et al., 2010), where experts may become *awake* or *asleep* at any given round. However, existing work on sleeping experts typically assumes a relatively small (fixed with respect to the time horizon $T$) pool of experts. This assumption is increasingly unrealistic in modern applications—such as Q&A platforms, social networks, or social proof marketing—where new experts emerge continuously. In such settings, the number of experts $K$ is often linear in the time horizon $T$ (*i.e.*, $K = \Theta(T)$), thus rendering the regret bounds achieved by existing algorithms for sleeping experts vacuous (see Section 1.3).

In this paper, we introduce and study the *Awakening Crowd of Experts* (ACE) problem, which adds a natural "awakening" structure to the sleeping experts problem, whereby a *new* expert awakens at each round, joins the pool, and remains available indefinitely. Thus, in our ACE problem, the learner must deal with a *huge* "crowd" of experts, as their total number $K$ equals the time horizon $T$. We show that this "awakening" structure fundamentally simplifies the problem, as it allows us to bypass the lower bounds for general sleeping experts and derive meaningful regret bounds even when $K = T$. To simplify the exposition and the analysis, we assume that a single new expert awakens at each round. However, most of our results extend to settings in which multiple experts awaken at each round.

Formally, in the ACE problem, at each round $t$ a new expert $i_t$ awakes and joins the pool of experts. Then, the learner has to pick which expert to follow among those in the pool of experts $\mathcal{S}_t = \{i_\tau\}_{\tau \in [t]}$ awaken at round $t$. As is customary in the literature on sleeping experts, we take as baseline the best ordering over experts $\sigma^\star$, and we measure the performance of the learner in terms of the regret against the policy that, at each round, selects the expert that is highest in such an ordering among the awake experts in $\mathcal{S}_t$.

### 1.1. Motivating Applications

*User Generated Content* (UGC) is nowadays ubiquitous in online platforms. Examples include social networks and similar platforms (*e.g.*, Reddit and StackOverflow) and online marketplaces (*e.g.*, Amazon and Facebook Market-

---
*Equal contribution, author order has been chosen randomly [1]Politecnico di Milano, Milan, Italy. Correspondence to: Anna Lunghi <anna.lunghi@polimi.it>, Gianmarco Genalti <gianmarco.genalti@polimi.it>.

*Proceedings of the 43 $^{rd}$ International Conference on Machine Learning*, Seoul, South Korea. PMLR 306, 2026. Copyright 2026 by the author(s).

place). Under this paradigm, a final user of the service is also a creator for the other users, including the upcoming new ones. In platforms such as Reddit, a user adds their own comment under a thread, possibly after having read all of the comments (or subthreads) generated by the other users before them. To promote user engagement, platforms like Reddit implement a peer feedback system, which usually consists of an *upvote* or a *like* that any user can assign to the content generated by another user. This system allows platforms to identify the users generating the highest quality content, and to promote their content in privileged slots or, in certain cases, to reward them with prizes or virtual badges. One can think of this application as an instance of the ACE problem. Indeed, the "awakening" structure is motivated by new users (or UGC) arriving sequentially, making it challenging to identify high-quality newcomers without disregarding proven veterans. Due to the large size of these platforms and their continual interaction mechanism, it is reasonable to assume that the number of experts can become huge and comparable to the number of decisions to make.

A similar model can be used in the context of *social proof marketing*, where UGC is specifically used as a leverage to increase click-through-rate or sales in online marketplaces. It is indeed common to see sellers and advertisers use good reviews from past users as a lead. Among the large number of past reviews and the new ones that continuously arrive, a seller/advertiser wants to identify the most convincing ones and to show them as frequently as possible.

Our ACE problem also captures settings involving *financial forecasting under regime shifts*, where the goal is to predict the direction of a stock price movement (*i.e.*, whether the price increases or not). It is common to assume that the forecaster has access to a pool of experts, where each expert $i$ predicts the direction of the price movement by looking $i$ days into the past. This naturally gives rise to a setting in which, at each day $t$, a new expert awakens—namely, the one relying on information from $t$ days earlier.

### 1.2. Summary of Contributions and Techniques

We investigate our ACE problem in both *stochastic* and *adversarial* regimes. These settings present fundamentally different challenges and require completely different approaches, as described in the following.

#### 1.2.1. STOCHASTIC ACE

In the stochastic setting, we establish a regret bound of order $\widetilde{\mathcal{O}}(T^{2/3})$. This result generalizes to the setting in which an arbitrary number of experts may awaken at each round, while maintaining a $\widetilde{\mathcal{O}}(T^{2/3} \log K)$ regret bound, where $K$ denotes the final number of awakened experts. Furthermore, we provide an instance-dependent analysis showing that the problem complexity is driven by the number of *switches*

made by a policy that follows the best ordering $\sigma^\star$, namely, the number of times such a policy changes the followed expert. We complement these upper bounds by providing two tight lower bounds.

Our algorithmic approach is based on an unusual *pessimism* in the face of uncertainty. Since a new expert awakens at every round, optimistic methods (like UCB) fail by overestimating the expected reward of new experts. Even simple empirical mean estimation is "too optimistic", as the lack of concentration for new experts leads to large overestimates. To address this, we use pessimism and lower confidence bounds. Then, the learner remains conservative, only switching to a newer expert when there is high confidence that its reward exceeds that of older experts. While this caution delays the adoption of better but newer experts, it prevents overestimating the reward of new experts.

We experimentally evaluate our algorithm by measuring its performance against the UCB algorithm on both real-world and simulated data. Our results show that UCB switches experts excessively often, whereas our algorithm successfully tracks the switches of an optimal policy. We provide the complete experimental evaluation in Appendix C.

#### 1.2.2. ADVERSARIAL ACE

The adversarial case is significantly more complex than the stochastic one. We first establish a fundamental lower bound: for any constant $\alpha \in (0, 1)$, the $\alpha$-regret is at least $\Omega(\alpha^2 T)$, effectively precluding standard sublinear regret upper bounds. This stems from a persistent tension between exploiting the optimal *old* expert and promptly switching to the best *new* expert.

To deal with trade-off described above, our algorithm randomizes between a weakly-adaptive regret minimizer—focused on following an optimal old expert over a sliding window $M$— and playing one of the newest $M$ arms at random. A major technical challenge is that the standard weakly-adaptive regret guarantee for a huge (linear in $T$) expert set scales with $\mathcal{O}(\sqrt{M \log T})$, which is not ideal for small intervals. We overcome this by deploying a meta-regret minimizer over a set of adaptive learners with varying window lengths. This construction reduces the dependency to $\mathcal{O}(\log \log T)$, yielding an $\alpha$-regret of order $\mathcal{O}(\sqrt{\log(\alpha^{-1})} \alpha^{3/2} T \sqrt{\log \log T})$. As a corollary, we achieve a $\widetilde{\mathcal{O}}(\log \log T)$ competitive ratio, provided that the reward achieved by the best ordering is linear in $T$.

### 1.3. Related Works

Our setting is a special case of the sleeping experts problem. Kleinberg et al. (2010) introduce this setting and provide optimal regret bounds for both the stochastic and adversarial cases. In both settings, the optimal regret is of order

$\widetilde{\mathcal{O}}(\sqrt{KT})$, where $K$ denotes the number of experts. Notice that, in our ACE problem, $K = T$, making this bound vacuous, *i.e.*, of order $\mathcal{O}(T)$. Subsequent works extend the setting in many different directions.

Another early work on sleeping experts is (Kanade et al., 2009). Differently from (Kleinberg et al., 2010)—and from our setting—the authors assume that the set of experts awake at a given round is selected stochastically. Interestingly, this assumption significantly simplifies the problem by reducing the dependence on $K$ in the regret bounds, allowing only a logarithmic dependence. Under the same setting, (Saha et al., 2020) propose the first efficient (*i.e.*, polynomial-time) algorithm achieving order-optimal $\mathcal{O}(\sqrt{T})$ regret.

In a previous work (Kanade & Steinke, 2014), the authors show that it is not possible to construct an efficient no-regret learner in the fully adversarial sleeping experts problem, *i.e.*, when both action sets and rewards are adversarially chosen. In (Emamjomeh-Zadeh et al., 2021), the authors shift the focus on $\alpha$-regret and provide a computationally efficient learners with constant $\alpha$-regret (w.r.t. to $T$), but scaling with $K$, making them unsuited when $K = T$.

Interestingly, (Shayestehmanesh et al., 2019) introduces the *dying experts* setting, a setting where experts can only go to sleep and never re-appear. In some sense, this setting is the opposite of ours; however, the authors assume that the initial number of experts $K$ is constant w.r.t. to $T$.

The awakening assumption appeared for the first time in (Ghalme et al., 2021), in the context of multi-armed bandits. The authors call their setting *ballooning multi-armed bandits*. However, there are some crucial differences between the two settings. First, we consider an expert setting, whereas they consider a bandit setting, which significantly changes the nature of the feedback. Second, they only focus on the stochastic version of the problem. Third, their algorithm relies on a set of assumptions regarding the arrival of the best arm. In particular, these assumptions allow for a bound on the time in which the last best action arrives, and never changes (with high probability) after that round. If the best arm is known to arrive soon enough, *e.g.*, in the round $t^* = o(T)$, then any bandit algorithm playing with the first $t^*$ actions can achieve sublinear regret. Differently, our instance-independent bounds are assumption free.

## 2. A Crowd of Awakening Experts

In this section, we introduce the *Awakening Crowd of Experts* problem, which is the focus of this paper. First, we provide some useful background on online learning with sleeping experts. Then, we formally define our problem and the associated learning goals. Finally, we discuss some existing results and their relation to ours.

### 2.1. Online Learning With Sleeping Experts

In the *online learning with experts* problem, a learner is faced with $T \in \mathbb{N}$ repeated decisions among a set of $K \in \mathbb{N}$ experts. At the beginning of each round $t \in [T]$,[1] every expert $i \in [K]$ privately generates a reward $X_{i,t}$, the learner selects an expert $I_t$ (possibly at random), and receives the corresponding reward $X_{I_t,t}$. After the decision, the full set of rewards $\{X_{i,t}\}_{i \in [K]}$ is revealed to the learner.

A relevant variant of this problem is the *sleeping experts* problem introduced by Kleinberg et al. (2010). In this setting, *not* all experts are available at every round. Instead, a sequence of sets $\mathcal{S}_t \subseteq [K]$ prescribes the experts available (or *awake*) at each round $t \in [T]$. After the learner takes a decision at round $t$, only the rewards of the experts awake at that round are revealed to the learner.

### 2.2. Awakening Crowd of Experts Problem

Next, we formally introduce our *Awakening Crowd of Experts* (ACE) problem. This is a special case of the sleeping experts problem, where there are $K = T$ possible experts and, at each round $t \in [T]$, a *new* expert $i_t \in [K]$ "awakens" by joining the set of available experts and never leaves it. Formally, this means that $\mathcal{S}_t := \mathcal{S}_{t-1} \cup \{i_t\}$ for every $t \in [T]$, where we let $\mathcal{S}_0 = \varnothing$ by convention. Thus, the sets of available experts $\mathcal{S}_t$ constitute a nested sequence.

Notice that the ACE problem begets considerable additional challenges compared to the classic sleeping experts problem, since the learner has to deal with a "crowd" of experts, as $K = T$. As discussed later in detail, this is extremely detrimental for existing algorithms for sleeping experts, where $K$ is usually assumed to be constant with respect to $T$. Indeed, the algorithms of Kleinberg et al. (2010) yield regret bounds that scale linearly in $T$ when $K = T$.

### 2.3. Learning Goal

We let $\mathcal{H}_t := \{I_1, \{X_{i,1}\}_{i \in \mathcal{S}_1}, \ldots, I_t, \{X_{i,t}\}_{i \in \mathcal{S}_t}\}$ be the history of the learner-environment interaction up to round $t \in [T]$. A learner's policy $\pi$ defines a (possibly randomized) mapping from the history $\mathcal{H}_{t-1}$ to a decision in $\mathcal{S}_t$ at every round $t \in [T]$. We denote by $\pi(t)$ the decision prescribed at round $t \in [T]$ by policy $\pi$.

As is customary in sleeping experts (Kleinberg et al., 2010), we evaluate the learner's performance in terms of regret against all possible orderings of the decisions. Let $\sigma$ be any ordering of the experts in $[K]$. Then, we define its corresponding *ordering policy* $\pi_\sigma$ as the policy that, at each round $t \in [T]$, prescribes the expert $\pi_\sigma(t) \in \mathcal{S}_t$ that is highest in the ordering $\sigma$. Then, the goal of a policy $\pi$ is to

---

[1]In this paper, we denote by $[N] := \{1, \ldots, N\}$ the set of the first $N \in \mathbb{N}$ natural numbers.

minimize its *expected ordering regret*, defined as

$$\mathbb{E}[R_T(\pi)] := \max_\sigma \sum_{t \in [T]} \mathbb{E}[X_{\pi_\sigma(t),t}] - \sum_{t \in [T]} \mathbb{E}[X_{\pi(t),t}],$$

where expectations are taken w.r.t. both the (possible) randomness of the environment and the (possible) randomness of $\pi$. In this paper, we also deal with a more general notion of regret called $\alpha$-regret. For $\alpha \in (0, 1)$, the *expected ordering $\alpha$-regret* of a policy $\pi$ is defined as follows:

$$\mathbb{E}[R_{\alpha,T}(\pi)] := \alpha \max_\sigma \sum_{t \in [T]} \mathbb{E}[X_{\pi_\sigma(t),t}] - \sum_{t \in [T]} \mathbb{E}[X_{\pi(t),t}].$$

We call $\frac{1}{\alpha}$ the *competitive ratio*, and we omit it when $\alpha = 1$ (in which case the definition collapses to standard regret). In the following, we write $I_t$ as the random expert sampled from $\pi(t)$, as the learner's policy $\pi$ will be clear from context. By letting $\sigma^\star \in \arg\max_\sigma \sum_{t \in [T]} \mathbb{E}[X_{\pi_\sigma(t),t}]$ be the best possible ordering of experts, we also let $i_t^\star := \pi_{\sigma^\star}(t)$ be the expert selected at round $t \in [T]$ by an optimal ordering policy. Then, we provide the following useful characterization of the structure of an optimal ordering policy. Intuitively, one such policy may select an expert $i_t$ only for a (possibly empty) sequence of consecutive rounds starting from the round $t$ in which the expert "awakens".

**Proposition 2.1.** *For every expert $i_{t_1}$ that "awakens" at some round $t_1 \in [T]$, either $\{t \in [T] \mid i_t^\star = i_{t_1}\} = \varnothing$ or there exists another round $t_2 \in [T] : t_2 \geq t_1$ such that:*

- *$i_t^\star = i_{t_1}$ for all $t \in [T] : t_1 \leq t \leq t_2$; and*
- *$i_t^\star \neq i_{t_1}$ for all $t \in [T] : t < t_1 \lor t > t_2$.*

*Proof.* Let $\sigma^\star$ denote the optimal ordering. To prove the proposition, we assume the existence of at least one round $t$ such that $i_t^\star = i_{t_1}$ and show that the required conditions are satisfied. First, observe that for all $t < t_1$, we must have $i_t^\star \neq i_{t_1}$ because $i_{t_1} \notin \mathcal{S}_t$ by definition of the awakening process.

To complete the proof, it suffices to show that if $i_{t_2}^\star = i_{t_1}$ for some $t_2 \geq t_1$, then $i_t^\star = i_{t_1}$ for all $t \in \{t_1 \dots, t_2\}$. This property follows from the fact that $i_{t_1}$ is the highest-ranked expert in $\sigma^\star$ among all experts in $\mathcal{S}_{t_2}$. Since the available sets are nested ($\mathcal{S}_t \subseteq \mathcal{S}_{t+1}$), $i_{t_1}$ remains the highest-ranked available expert for any $t \in \{t_1 \dots, t_2\}$. □

We study both the *stochastic* and the *adversarial* variants of the ACE problem, which we introduce in the following.

**Stochastic ACE**    In the *stochastic* ACE problem, the rewards generated by the experts are i.i.d. random variables. In particular, for every expert $i \in [K]$, there exists a probability distribution $\nu_i$ with support bounded in $[0, 1]$ and fixed mean $\mu_i := \mathbb{E}_{\nu_i}[X_{i,t}]$ for every $t \in [T]$. We let $\mathcal{S}_T^\star \subseteq [K]$

be the subset of experts that are picked by an optimal ordering policy for at least one of the $T$ rounds. Then, given any index $j = 1, \ldots, |\mathcal{S}_T^\star|$, we denote with $\mathcal{S}_T^\star(j)$ the $j$-th expert in $\mathcal{S}_T^\star$ according to the order in which the experts "awaken". Finally, we let $\Upsilon^\star := |\mathcal{S}_T^\star|$ be the number of *switches* that an optimal ordering policy needs to perform. This quantity plays a key role in our regret guarantees.

**Adversarial ACE**    In the *adversarial* ACE problem, the rewards are generated by each expert arbitrarily. In particular, for every expert $i_t$ "awakening" at round $t \in [T]$, there exists a fixed sequence of rewards $\{X_{i_t,\ell}\}_{\ell=t}^T$.

### 2.4. Existing Results

As previously discussed, our setting is a special case of the sleeping experts problem. In (Kleinberg et al., 2010), the authors provide algorithms and regret bounds for both the stochastic and the adversarial sleeping experts problem. For a set of $K$ experts able to freely awake and go back to sleep, the FTAL algorithm suffers a regret that can be upper bounded as $\mathcal{O}(\sqrt{KT})$ in the stochastic sleeping experts. The Hedge algorithm suffers a regret that can be upper bounded as $\mathcal{O}(\sqrt{K \ln KT})$ in the adversarial sleeping experts. These bounds are tight, as they match the lower bounds provided in (Kleinberg et al., 2010). Setting $K = T$ would result in vacuous bounds, *i.e.*, $\mathcal{O}(T)$.

## 3. Stochastic Awakening Crowd of Experts

In this section, we focus on the *stochastic* version of our problem. Our main result states that it is possible to obtain sublinear regret even when the number of experts is $T$. The assumption that the experts are only "awakening" plays a key role in our result, since in standard sleeping experts a lower bound of the order of $\Omega(T)$ can be derived from Lemma 5 and Lemma 6 in (Kleinberg et al., 2010).

### 3.1. A Lower Bound on the Regret

First, we provide an impossibility result establishing the best possible regret upper bound that any algorithm can achieve in the stochastic ACE problem. We give a characterization based on the number of *switches* among experts made by an optimal ordering policy, namely $\Upsilon^\star$. Formally, we can lower bound the regret as follows:

**Proposition 3.1** (Lower Bounds for Stochastic ACE)**.** *In the ACE problem, for any policy $\pi$, any $T \in \mathbb{N}$, and any number of switches $\Upsilon^\star \leq T^{1/3}$, there exists an instance in which the expected ordering regret can be lower bounded as*

$$\mathbb{E}[R_T(\pi)] \geq \Omega\left(\sqrt{\Upsilon^\star T}\right). \tag{1}$$

*Moreover, for any policy $\pi$ and $T \in \mathbb{N}$, there exists an instance in which the expected ordering regret can be lower*

*bounded as*

$$\mathbb{E}[R_T(\pi)] \geq \Omega\left(T^{2/3}\right). \qquad (2)$$

*Proof Sketch.* The instance-independent result is proven by constructing a set of instances in which the trial is partitioned into $T^{1/3}$ epochs of length $T^{2/3}$. At the beginning of every epoch, two experts awake: one has a slightly better expected reward then the previous ones, and becomes the new optimal expert; the other is identical to the optimal expert of the previous epoch. All of the other experts awakening during an epoch provide no reward at all. The learner is aware of the instance construction but does not distinguish the identity of the first two experts of an epoch. In fact, every epoch resorts to a standard stochastic two-expert problem, and the regret can be lower bounded as $\Omega(\sqrt{T^{2/3}})$. Since no information can be carried from one epoch to the next one, the final regret is just the sum of the regrets on all epoch, *i.e.*, $\Omega(T^{2/3})$. One interesting observation is that the optimal epoch size is exactly $T^{1/3}$: a larger number of epochs would make the increase in reward between two successive epochs smaller, decreasing the cumulative regret. A complete proof can be found in Appendix A. $\square$

### 3.2. A Sublinear Upper Bound on the Regret

Next, we present a simple algorithm for the stochastic ACE problem. The algorithm employs a *pessimistic* policy $\pi_{LCB}$, whose pseudo-code is reported in Algorithm 1.

---
**Algorithm 1** Pessimistic Policy $\pi_{LCB}$

---
**Require:** Time horizon $T \in \mathbb{N}$
 1: **for** $t = 1, \ldots, T$ **do**
 2:      Compute $LCB_{i,t} = \widehat{\mu}_{i,t} - \sqrt{\frac{6 \ln T}{n_{i,t}}}$ for all $i \in \mathcal{S}_{t-1}$
 3:      Choose expert $I_t \in \arg\max_{i \in \mathcal{S}_{t-1}} LCB_{i,t}$
 4:      Receive $\{X_{i,t}\}_{i \in \mathcal{S}_t}$
 5: **end for**

---

At each round $t \in [T]$, the algorithm computes the *Lower Confidence Bound* (LCB) for every available expert $i \in \mathcal{S}_{t-1}$ (Line 2), as follows:

$$LCB_{i,t} = \widehat{\mu}_{i,t} - \sqrt{\frac{6 \ln k_T}{n_{i,t}}},$$

where $n_{i,t} := \sum_{\ell=1}^{t-1} \mathbb{1}_{\{i \in \mathcal{S}_\ell\}}$ is the number of rounds in which expert $i$ has been "awake" up to round $t-1$ and $\widehat{\mu}_{i,t} := \frac{1}{n_{i,t}} \sum_{\ell=t-1-n_{i,t}}^{t-1} X_{i,\ell}$ is the empirical mean of the observed rewards for expert $i$. Then, at round $t \in [T]$, the algorithm selects the expert $I_t \in \arg\max_{i \in \mathcal{S}_{t-1}} LCB_{i,t}$ (Line 3). Finally, the algorithm observes the rewards generated by the "awake" experts (Line 4).

**Theorem 3.2** (Upper Bounds for Stochastic ACE). *In the ACE problem, the policy $\pi_{LCB}$ suffers an expected ordering regret upper bounded as*

$$\mathbb{E}[R_T(\pi_{LCB})] \leq \widetilde{\mathcal{O}}\left(\sqrt{\Upsilon^\star T}\right). \qquad (3)$$

*Moreover, the policy $\pi_{LCB}$ suffers a worst-case expected ordering regret upper bounded as*

$$\mathbb{E}[R_T(\pi_{LCB})] \leq \widetilde{\mathcal{O}}\left(T^{2/3}\right). \qquad (4)$$

*Proof Sketch.* Here, we focus on the instance-independent bound. Our analysis is based on a simple charging scheme: we account the optimal expert at time $t$ to the expert chosen at time $t + T^{2/3}$, which we call $\tilde{I}_t$.

In what follows, to simplify the exposition, we assume that the high-probability event (occurring with probability at least $1-\delta$) under which all estimates concentrate is satisfied. Hence:

$$|\mu_i - \widehat{\mu}_{i,t}| \leq \sqrt{\frac{2\ln(\delta^{-1})}{n_{i,t}}}$$

for any expert $i$ and round $t$. Then, we get that, for each $t \leq T - T^{2/3}$,

$$\mu_{i_t^\star} - \mu_{\widetilde{I}_t} = \mu_{i_t^\star} - \mu_{\widetilde{I}_t} \pm LCB_{\widetilde{I}_t, t+T^{2/3}} \pm LCB_{i_t^\star, t+T^{2/3}}$$
$$\leq \mu_{i_t^\star} - LCB_{i_t^\star, t+T^{2/3}}$$
$$\leq \sqrt{\frac{6\ln T}{T^{2/3}}}.$$

Hence, the final regret can be bounded by noting that

$$\sum_{t \in [T]} \left(\mu_{i_t^\star} - \mu_{I_t}\right) \leq \sum_{t \in [T-T^{2/3}]} \left(\mu_{i_t^\star} - \mu_{\widetilde{I}_t}\right) + T^{2/3}$$
$$= T^{2/3}\left(2\sqrt{6\ln T} + 1\right).$$

A complete proof and the proof of the instance-dependent bound can be found in Appendix A. $\square$

*Remark* 3.3 (Extension to an arbitrarily large crowd). The upper bound on the regret for $\pi_{LCB}$ continues to hold—up to minor modification in the algorithm—in a more general setting than the ACE problem. Indeed, if we allow for multiple experts to "wake up" at every round, then the bounds remain valid. In this general case, where the final number of experts $K$ may be larger than $T$, we only need to modify how the LCB is computed, modifying $\ln T$ with $\ln K$ to take into account the larger number of random variables that should concentrate. This modification only impacts a logarithmic term, and not the dominating terms.

# 4. Adversarial Awakening Crowd of Experts

In this section, we address $\alpha$-regret minimization in the *adversarial* version of our problem. Interestingly, in the adversarial setting, the assumption that experts can only awaken is *not* sufficient to ensure sublinear regret. In particular, in Section 4.1 we show that, for any constant $\alpha \in (0, 1)$, this assumption is *not* enough to even obtain sublinear $\alpha$-regret. In Section 4.2, we complement this result with an almost matching upper bound on the $\alpha$-regret.

## 4.1. Linear $\alpha$-Regret is Unavoidable

First, we show that there exists a crucial trade-off between the competitive ratio and the $\alpha$-regret: sublinear $\alpha$-regret cannot be attained while keeping $\alpha$ constant and independent of $T$. Formally:

**Proposition 4.1** (Lower Bound for Adversarial ACE). *Let $\alpha \in (0, 1)$. Then, for any policy $\pi$ and any $T \in \mathbb{N}$, there exists an instance in which the expected ordering $\alpha$-regret can be lower bounded as*

$$\mathbb{E}[R_{\alpha,T}(\pi)] \geq \frac{1}{4}\alpha^2 T.$$

*Proof Sketch.* The construction is inspired by the one used in the lower bound for the stochastic ACE problem. Specifically, we partition the time horizon into $T/M$ epochs of length $M$, where $M$ is a parameter determined by $\alpha$. The key difference from the stochastic case lies in the use of adversarial rewards to strengthen the lower bound. In the proof of Proposition 3.1, rewards must increase across successive epochs to prevent the learner from exploiting profitable experts from previous epochs. In the adversarial construction, instead, we can set the rewards of any expert that awoke in past epochs to zero, effectively removing this constraint.

Formally, within each epoch $\{t_1, \ldots, t_2\}$, we set all rewards to zero for the first $M-1$ rounds. In the final round $t_2$, a single expert—chosen uniformly at random from the set of experts that awoke during that epoch (*i.e.*, in $\mathcal{S}_{t_2} \setminus \mathcal{S}_{t_1-1}$)—is assigned a reward of 1. All other experts receive reward zero in this round as well.

Under this construction, the best a learner can do at round $t_2$ is to select an expert uniformly at random from those that awoke during that epoch, yielding an expected reward of $1/M$. In contrast, the policy induced by the best ordering selects the optimal expert for each epoch and receives a reward of 1. Summing over all $T/M$ epochs, the learner's total expected reward is $T/M^2$, whereas the optimal ordering achieves a reward of $T/M$. Setting $M = 2/\alpha$ and rearranging terms yields the desired lower bound. A complete proof is provided in Appendix B. $\square$

The regime in which the cumulative reward of an optimal ordering policy $\pi_{\sigma^\star}$ is $\Theta(T)$ is particularly relevant and

meaningful: one can re-arrange a bound over the $\alpha$-regret as a bound on the competitive ratio only, *i.e.*,

$$\sum_{t \in [T]} \mathbb{E}[X_{\pi(t),t}] \leq \left(\alpha - c\frac{\alpha^2}{4}\right) \max_\sigma \sum_{t \in [T]} \mathbb{E}[X_{\pi_\sigma(t),t}],$$

where $c > 0$ is a universal constant.

A reasonable algorithm for adversarial ACE must then be able to balance the trade-off between $\alpha$ and $R_{\alpha,T}$. This is particularly true when the cumulative reward of an optimal ordering policy is linear in $T$ and the competitive ratio has a leading role over the additive regret.

## 4.2. An Almost Matching $\alpha$-Regret Bound

Now, we present an algorithm for the adversarial ACE problem. Our algorithm randomizes between two different regret minimizers. This is necessary for the following reasons.

### 4.2.1. WHY TWO SEPARATE STRATEGIES?

By leveraging Proposition 2.1, the policy induced by the best ordering $\sigma^\star$ can be interpreted as a partition of the time horizon $[T]$ into a collection of disjoint intervals $\mathcal{I}^\star$. For each interval $I = \{t_1, \ldots, t_2\} \in \mathcal{I}^\star$, an optimal ordering policy consistently selects expert $t_1$ throughout the interval.

This observation suggests that distinct strategies are required depending on the length of the intervals. In particular, employing the same regret-minimization procedure uniformly across all interval lengths is suboptimal, and can be especially detrimental when intervals are very short. Indeed, the length of an interval $I \in \mathcal{I}^\star$ determines whether an additive regret guarantee or a multiplicative competitive ratio guarantee is more appropriate and easier to achieve.

Specifically, for a given threshold $M$, we define $\mathcal{I}^\star_{\text{long}} \subseteq \mathcal{I}^\star$ as the set of intervals with length at least $M$, and let $\mathcal{I}^\star_{\text{short}} = \mathcal{I}^\star \setminus \mathcal{I}^\star_{\text{long}}$ denote the remaining short intervals. The general idea is that when $I \in \mathcal{I}^\star_{\text{short}}$, the number of plausible optimal experts is very limited—at most $M$—which makes achieving a competitive ratio of $M$ straightforward. Conversely, when $I \in \mathcal{I}^\star_{\text{long}}$, the interval is sufficiently long to allow learning, and it is therefore natural to aim for a small additive regret.

### 4.2.2. HIGH-LEVEL CONSTRUCTION

We propose an algorithm (whose pseudo-code is in Algorithm 2) parameterized by $\alpha \in (0, 1)$, which defines the target competitive ratio. The algorithm first determines the window size $M = \frac{1}{\alpha} - 1$ (Line 1) and the probabilities $p_1 = \frac{M}{M+1}$ and $p_2 = 1 - p_1$ (Line 2). Then, as also shown in Figure 1, the learner adopts a randomized strategy: with probability $p_1$, it samples an expert uniformly at random from the $M$ most recently awakened experts,

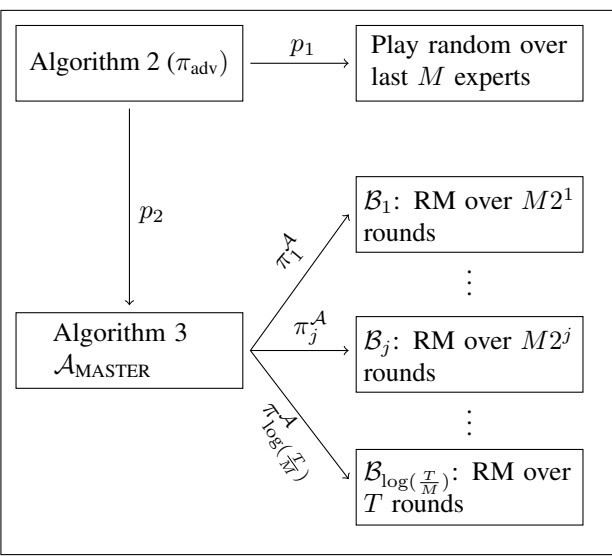

*Figure 1.* Structure of the algorithm for adversarial ACE.

---

**Algorithm 2** Policy for Adversarial ACE ($\pi_{\text{adv}}$)

---

**Require:** Parameter $\alpha \in (0, 1)$
1: $M \leftarrow \frac{1}{\alpha} - 1$
2: $p_1 \leftarrow \frac{M}{M+1}, p_2 \leftarrow 1 - p_1$
3: **for** $t \in [T]$ **do**
4:      with probability $p_1$: $\pi_t \sim \mathcal{U}(\mathcal{S}_t \setminus \mathcal{S}_{t-M})$
5:      with probability $p_2$: pick $\pi_t$ according to $\mathcal{A}_{\text{MASTER}}$
6: **end for**

---

*i.e.*, $\pi_t \sim \mathcal{U}(\mathcal{S}_t \setminus \mathcal{S}_{t-M})$. This random exploration ensures robust performance over short intervals, where selecting a recent expert is likely to yield good rewards. Conversely, with probability $p_2$, the algorithm follows a dedicated regret minimizer (specified in Algorithm 3) designed to optimize performance over sufficiently long intervals.

### 4.2.3. HANDLING SHORT INTERVALS

Consider an interval $I \in \mathcal{I}^\star_{\text{short}}$, *i.e.*, an interval such that $|I| \leq M$. This immediately implies that $i^\star_t \in \mathcal{S}_t \setminus \mathcal{S}_{t-M}$ for all $t \in I$. Hence, sampling an expert uniformly at random from the $M$ most recently awakened ones—a strategy we execute with probability $p_1$—attains a competitive ratio of $M$ over the interval $I$.

### 4.2.4. HANDLING LONG INTERVALS

For intervals $I \in \mathcal{I}^\star_{\text{long}}$, *i.e.*, such that $|I| > M$, our main idea is to exploit the interval length to learn an (approximately) optimal expert over the interval. We achieve this by using a regret minimizer $\mathcal{A}_{\text{MASTER}}$ (see Algorithm 3). To guarantee that $\mathcal{A}_{\text{MASTER}}$ provides small regret on *intervals*, it must satisfy a *weakly-adaptive* regret guarantee (according to the definition given by (Castiglioni et al., 2024) and

originally introduced in (Hazan & Seshadhri, 2007)). In particular, it must ensure low regret (*i.e.*, sublinear in $T$) over any sub-interval. However, standard adaptive algorithms face a significant hurdle: the massive expert pool introduces a $\log(T)$ multiplicative overhead in the regret. Given that the number of intervals $T/M$ could be high, this dependence might severely degrade the performance. To mitigate this, we employ a meta-regret minimizer $\mathcal{A}$ that dynamically adapts to the unknown length of the current interval $I$. In particular, the meta-regret minimizer maintains a set of base regret minimizers $\{\mathcal{B}_j\}_{j \in [\log(T/M)]}$.

At each round $t$, the meta-algorithm $\mathcal{A}$ selects a candidate $\mathcal{B}_{j_t}$ to minimize the adaptive regret relative to the best-performing expert in the current interval. By employing a standard regret minimizer such as Fixed Share (Cesa-Bianchi et al., 2012), we obtain the following guarantee for any interval $\{t_1, \ldots, t_2\}$ of length at most $M$:

$$\mathbb{E}\left[ \max_{j \in [\log(T/M)]} \sum_{t \in \{t_1, \ldots, t_2\}} r_t(\mathcal{B}_j) - r_t(\mathcal{A}_t) \right]$$
$$\leq \mathcal{O}(\sqrt{M \log(M \log(T))}),$$

where $r_t(\mathcal{A}_t)$ and $r_t(\mathcal{B}_j)$ denote the rewards of the meta-regret minimizer and the $j$-th base regret minimizer at round $t$, respectively. As shown in the following, this hierarchical structure effectively reduces the terms due to the total number of experts from $\log(T)$ to $\log(\log T)$.

### 4.2.5. IMPLEMENTING THE BASE MINIMIZERS

We implement the base regret minimizers $\{\mathcal{B}_j\}_{j \in [\log(T/M)]}$ using a doubling scheme, where each $\mathcal{B}_j$ is optimized for intervals of length approximately $M \cdot 2^j$. This approach allows us to minimize the expert pool of each regret minimizer by resetting $\mathcal{B}_j$ every $M \cdot 2^j$ rounds. Specifically, given two rounds $t_1 = qM2^j + 1$ and $t_2 = (q+1)M2^j$ for a $q \in \mathbb{N}$, the regret minimizer $\mathcal{B}_j$ works only with the expert set $\mathcal{S}_{t_2} \setminus \mathcal{S}_{t_1 - M2^j}$.[2] We require $\mathcal{B}_j$ to satisfy a weakly-adaptive regret guarantee over the interval $\{t_1, \ldots, t_2\}$: for any sub-interval $\{t', \ldots, t''\} \subseteq \{t_1, \ldots, t_2\}$, it holds that

$$\mathbb{E}\left[ \max_{a \in \mathcal{S}_{t_2} \setminus \mathcal{S}_{t_1 - M2^j}} \sum_{t \in \{t', \ldots, t''\}} (X_{a,t} - r_t(\mathcal{B}_j)) \right]$$
$$\leq \mathcal{O}\left( \sqrt{(t_2 - t_1) \log(t_2 - t_1)} \right),$$

where we use the fact that $|\mathcal{S}_{t_2} \setminus \mathcal{S}_{t_1 - M2^j}| = \mathcal{O}(t_2 - t_1)$.[3]

---

[2]The inclusion of experts from $\mathcal{S}_{t_1-1} \setminus \mathcal{S}_{t_1 - M2^j}$ ensures that the optimal expert in an interval $I$ of length $M$ remains within the learner's active pool, even when the interval $I \in \mathcal{I}^\star_{\text{long}}$ is misaligned with $\{t_1, \ldots, t_2\}$. This "buffer" guarantees that the optimal expert is available till round $t_2$.

[3]Note that at round $t$, the learner cannot select experts in $\mathcal{S}_{t_2} \setminus \mathcal{S}_t$ as they have not yet awakened. This is easily addressed by: (i)

---

**Algorithm 3** Regret minimizer $\mathcal{A}_{\text{MASTER}}$

---

**Require:** Interval length $M$
1: **for** $t \in [T]$ **do**
2:     **for** $j \in [\log(T/M)]$ **do**
3:         **if** $t-1$ is a multiple of $M2^j$ **then**
4:             Reset $\mathcal{B}_j$ with experts $\mathcal{S}_{t+M2^j} \setminus \mathcal{S}_{t-M2^j}$
5:         **end if**
6:     **end for**
7:     Sample RM $\mathcal{B}_{j_t}$ according to RM $\mathcal{A}$
8:     Sample expert $I_t$ according to RM $\mathcal{B}_{j_t}$
9: **end for**

---

This fine-grained optimization of the expert pool is essential to optimize the bound. The intuition is that while longer intervals ($2^j M$) naturally involve a larger pool of experts—and thus a larger additive regret term—this cost is better absorbed over a longer time horizon, maintaining a good competitive ratio.

### 4.2.6. $\alpha$-REGRET GUARANTEES

Now, we provide the guarantees of Algorithm 2.

**Theorem 4.2** (Upper Bound for ACE). *For any $\alpha \in [0,1]$, Algorithm 2 guarantees*

$$R_{\alpha,T}(\pi_{\text{adv}}) \leq \mathcal{O}\left(\sqrt{\log(\alpha^{-1})}\alpha^{3/2}T\sqrt{\log(\log(T))}\right).$$

*Proof Sketch.* We decompose the expected reward of $\pi_{\text{adv}}$ into two components: one for short intervals and one for long ones. Formally:

$$\sum_{t=1}^{T}\mathbb{E}[X_{\pi_{\text{adv}}(t),t}] =$$

$$\sum_{I \in \mathcal{I}^\star_{\text{long}}}\sum_{t \in I}\mathbb{E}[X_{\pi_{\text{adv}}(t),t}] + \sum_{I \in \mathcal{I}^\star_{\text{short}}}\sum_{t \in I}\mathbb{E}[X_{\pi_{\text{adv}}(t),t}].$$

For the component regarding short intervals, we observe that $\pi_{\text{adv}}$ chooses the random regret minimizer with probability $p_1$ and this regret minimizer picks the right expert with probability at least $\frac{1}{M}$. Hence, we get:

$$\sum_{I \in \mathcal{I}^\star_{\text{short}}}\sum_{t \in I}\mathbb{E}[X_{\pi_{\text{adv}}(t),t}] \geq p_1\frac{1}{M}\sum_{I \in \mathcal{I}^\star_{\text{short}}}\sum_{t \in I}\mathbb{E}[X_{i^\star_t,t}].$$

For the component regarding long intervals, we exploit the weakly-adaptive regret guarantees. Consider an interval $I \in \mathcal{I}^\star_{\text{long}}$. Informally, we have that the regret minimizer $\mathcal{A}_{\text{MASTER}}$ has no-regret with respect to following the regret minimizer $B_{j^\star_I}$ that "guesses" the right interval length, and

---

assigning a reward of 0 to unavailable experts, and (ii) selecting a random awake expert if the minimizer chooses an unavailable one.

---

$B_{j^\star_I}$ has no-regret in this interval (see Appendix B for a more formal analysis). Hence, we get

$$\sum_{I \in \mathcal{I}^\star_{\text{long}}}\sum_{t \in I}\left(\mathbb{E}[r_t(\mathcal{A}_{\text{MASTER}})] - \mathbb{E}[X_{i^\star_t,t}]\right)$$

$$\leq \mathcal{O}\left(\frac{T}{\sqrt{M}}\sqrt{\log(M\log(T))}\right).$$

Combining these results we obtain that $\pi_{\text{adv}}$ gets expected reward at least

$$p_1\frac{1}{M}\sum_{I \in \mathcal{I}^\star_{\text{short}}}\sum_{t \in I}\mathbb{E}[X_{i^\star_t,t}]$$

$$+ p_2\left[\sum_{I \in \mathcal{I}^\star_{\text{long}}}\sum_{t \in I}\mathbb{E}[X_{i^\star_t,t}] - \mathcal{O}\left(\frac{T}{\sqrt{M}}\sqrt{\log(M\log(T))}\right)\right].$$

Setting $p_1 = \frac{M}{M+1}, p_2 = \frac{1}{M+1}$, we get:

$$\frac{1}{M+1}\left(\sum_{t=1}^{T}\mathbb{E}[X_{\pi_{\sigma^\star}(t),t}]\right)$$

$$- \mathcal{O}\left(\frac{1}{M+1}\frac{T\sqrt{\log(M\log(T))}}{\sqrt{M}}\right).$$

Setting $\alpha = \frac{1}{1+M}$ we get the desired guarantees. A complete proof is provided in Appendix B. $\square$

As a corollary, we achieve a $\tilde{\mathcal{O}}(\log\log T)$ competitive ratio under the assumption that the optimal reward grows linearly in $T$. To obtain this result, we require a rough estimate of the optimal reward, which allows us to tune $\alpha$ as a function of this estimate. Formally:

**Corollary 4.3** (Competitive Ratio). *Suppose $OPT = \max_\sigma \sum_{t \in [T]} X_{\pi_\sigma(t),t} \geq \beta T$ for an absolute constant $\beta$ known to the learner. Then, Algorithm 2 guarantees reward at least*

$$\widetilde{\Omega}\left(\frac{1}{\log\log(T)}\right)OPT,$$

*where we use the notation $\widetilde{\Omega}(\cdot)$ to hide polylogarithmic terms in its argument.*

*Proof.* The crucial intuition behind the proof is that the multiplicative approximation, *i.e.*, $\alpha$, decreases more slowly than the additive regret, which depends on $\alpha^{3/2}$. Hence, for $\alpha$ small enough, the multiplicative term $\alpha OPT$ "absorbs" the additive regret.

Clearly, the term $OPT$ also plays a role in this trade-off, which is negligible only when $OPT = \Omega(T)$.

Formally, it is sufficient to apply Theorem 4.2 with $\alpha = \widetilde{\Omega}(\frac{1}{(\log\log(T))})$, where we highlight that the constant $\beta$ is

hidden in the $\widetilde{\Omega}(\cdot)$ notation. Then, we get

$$\alpha \, OPT - \mathcal{O}\left(\log(\alpha^{-1})\alpha^{3/2}T\log(\log(T))\right)$$
$$\geq \frac{\alpha}{2}OPT \geq \widetilde{\Omega}\left(\frac{1}{\log\log(T)}\right)OPT.$$

This concludes the proof. $\qquad\qquad\qquad\square$

## 5. Open Problem: Optimal Bounds With Adversarial Rewards

We conclude the paper leaving an intriguing open problem. While we provide nearly matching upper and lower bounds on the $\alpha$-regret in adversarial settings, a small gap remains. Specifically, our upper bound incurs an additional $\log\log T$ factor, which arises from the hierarchical nature of our meta-learner $\mathcal{A}_{\text{MASTER}}$ (Algorithm 3), as it balances a logarithmic number of sub-regret minimizers.

Beyond refining the dependency on $\alpha$, it remains an open question whether this $\log\log T$ term is required or suboptimal and due to our doubling scheme. In particular:

*Is it possible to achieve expected ordering $\alpha$-regret $\mathcal{O}(\alpha^c T)$ for a constant $c > 1$?*

We leave closing this gap as an intriguing future work.

## Acknowledgements

This publication was funded by the EU Horizon project ELIAS (European Lighthouse of AI for Sustainability, No. 101120237).

## Impact Statement

This paper presents work whose goal is to advance the field of Machine Learning. There are many potential societal consequences of our work, none which we feel must be specifically highlighted here.

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

# A. Omitted Proofs for the Stochastic ACE Problem

**Lemma A.1** (Lemma 6 from (Kleinberg et al., 2010)). *Let $T > 0$ and $\delta_T \geq \Omega\left(\frac{1}{\sqrt{T}}\right)$. Let $1 > \mu_1 > \mu_2 > 0$ be such that $\mu_1 - \mu_2 = \delta_T$. Then, there exists an expert problem instance with two experts with expected reward $\mu_1$ and $\mu_2$ such that, for large enough $T$, the regret of any policy $\pi$ is lower bounded as*

$$\mathbb{E}[R_T(\pi)] \geq \Omega(\delta_T^{-1}),$$

*as far as $\mu_1$ and $\mu_2$ are bounded away from 0.*

**Proposition 3.1** (Lower Bounds for Stochastic ACE). *In the ACE problem, for any policy $\pi$, any $T \in \mathbb{N}$, and any number of switches $\Upsilon^\star \leq T^{1/3}$, there exists an instance in which the expected ordering regret can be lower bounded as*

$$\mathbb{E}[R_T(\pi)] \geq \Omega\left(\sqrt{\Upsilon^\star T}\right). \tag{1}$$

*Moreover, for any policy $\pi$ and $T \in \mathbb{N}$, there exists an instance in which the expected ordering regret can be lower bounded as*

$$\mathbb{E}[R_T(\pi)] \geq \Omega\left(T^{2/3}\right). \tag{2}$$

*Proof.* The results can be proven by reducing the problem to a sequence of standard stochastic expert problems. Accordingly, for the remainder of the proof we consider instances in which $X_{i,t}$ is drawn from a Bernoulli distribution with parameter $\mu_i$. Let $\Upsilon^\star$ denote the number of switches. We construct a family of instances as follows.

We partition the set of times into equal length epochs $E_j = \{tj + 1, \ldots, t(j+1)\}$ such that $|E_j| = L = \frac{T}{\Upsilon^\star}$ for every $j \in [\Upsilon^\star]$ (assume for simplicity that $\Upsilon^\star$ exactly divides $T$). In each epoch $E_j$, any expert $i$ that arrives after the second round is characterized by $\mu_i = 0$, while the first two experts to arrive in the epoch, denoted by $i_{jL+1}$ and $i_{jL+2}$, are characterized by parameters $\frac{1}{2} + j\epsilon$ and $\frac{1}{2} + (j-1)\epsilon$, chosen in some random order.

More formally, we can build a family of $2^{\Upsilon^\star}$ instances, one for each binary sequence $b = \{b_j\}_{j=0}^{\Upsilon^\star - 1}$ with $b_j \in \{0, 1\}$, such that in the instance $b$ the rewards are defines as

$$\mu_i^b = \begin{cases} \frac{1}{2} + j\epsilon & \text{if } (i = Lj + 1 \wedge b_j = 0) \vee (i = Lj + 2 \wedge b_j = 1) \\ \frac{1}{2} + (j-1)\epsilon & \text{if } (i = Lj + 1 \wedge b_j = 1) \vee (i = Lj + 2 \wedge b_j = 0) \\ 0 & \text{otherwise.} \end{cases}$$

To guarantee that the instances are well defined, *i.e.*, $\mu_i \in [0, 1]$, it must be that $\frac{1}{2} + j\epsilon \leq 1$ for all $j \in \{0, \ldots, \Upsilon^\star - 1\}$, which is satisfied by $\epsilon \leq \frac{1}{2\Upsilon^\star}$.

First, notice that, knowing the specific instance $b$, it is possible to define an ordering $\sigma^\star$ that in each epoch $E_j$, from round $Lj + 1$ to $L(j+1)$, selects $i^\star$ such that $\mu_{i^\star} = \frac{1}{2} + j\epsilon$. Instead, an algorithm that does not know in advance which instance $b$ it is facing essentially has to identify the best expert of each epoch $j$ among the first two experts in that epoch, *i.e.*, in each epoch $j$ it has to correctly identify $b_j$.

In particular, by construction, for any algorithm there exists an algorithm at least as good but that for any $t \in E_j$ restricts its expert set to $\{i_{Lj+1}, i_{Lj+2}\}$. Indeed,

$$\min\{\mu_{jL+1}, \mu_{jL+2}\} = \frac{1}{2} + (j-1)\epsilon \geq \mu_i \quad \forall t \in E_j, i \in \mathcal{S}_t.$$

Hence, for this family of instances it cannot exist an algorithm that performs better than one that faces $\Upsilon^\star$ separate standard stochastic expert problems between two experts, where the optimal expert has a reward $\epsilon$ greater than that of the other, and time horizon $L - 1$.

We can then use Lemma A.1 to lower bound the regret in epoch $j$ with $\Omega\left(\frac{1}{\epsilon}\right)$. We can obtain the final bound by summing over the epochs, and letting $\epsilon = \frac{1}{4}\sqrt{\frac{\Upsilon^\star}{T}}$. Formally:

$$\mathbb{E}[R_T(\pi)] \geq \sum_{j=1}^{\Upsilon^\star} \Omega\left(\frac{1}{\epsilon}\right) = \Omega\left(\Upsilon^\star \sqrt{\frac{T}{\Upsilon^\star}}\right) = \Omega\left(\sqrt{\Upsilon^\star T}\right).$$

To conclude, for the choice $\epsilon = \frac{1}{4}\sqrt{\frac{\Upsilon^\star}{T}}$, the instances are well defined, *i.e.*, $\epsilon \leq \frac{2}{\Upsilon^\star}$, only as long as $\Upsilon^\star \leq 64T^{1/3}$.

In particular, letting $\Upsilon^\star = \mathcal{O}(T^{1/3})$:

$$\mathbb{E}[R_T(\pi)] \geq \Omega(T^{2/3}).$$

This concludes the proof. $\qquad\qquad\qquad\qquad\qquad\qquad\qquad\qquad\qquad\qquad\qquad\qquad\qquad\qquad\qquad\qquad\quad\square$

**Theorem 3.2** (Upper Bounds for Stochastic ACE). *In the ACE problem, the policy $\pi_{LCB}$ suffers an expected ordering regret upper bounded as*

$$\mathbb{E}[R_T(\pi_{LCB})] \leq \widetilde{\mathcal{O}}\left(\sqrt{\Upsilon^\star T}\right). \tag{3}$$

*Moreover, the policy $\pi_{LCB}$ suffers a worst-case expected ordering regret upper bounded as*

$$\mathbb{E}[R_T(\pi_{LCB})] \leq \widetilde{\mathcal{O}}\left(T^{2/3}\right). \tag{4}$$

*Proof.* We divide the trial into epochs. Epoch $\mathcal{I}_j := \{t \in [T] : \mu_{\mathcal{S}_T^\star(j)} > \mu_i \;\; \forall i \in \mathcal{S}_t\}$ corresponds to the set of time steps in which the best available expert is the $j$-th element of the sequence $\mathcal{S}_T^\star$. Also, let $L_j := |\mathcal{I}_j|$.

Let $\delta \in (0, 1)$. We introduce the *good event* $\mathcal{E}_T(\delta)$, defined as

$$\mathcal{E}_T(\delta) := \left\{ |\mu_i - \widehat{\mu}_{i,t}| \leq \sqrt{\frac{2\ln\delta^{-1}}{n_{i,t}}} \quad \forall t \in [T], \;\; \forall i \in \mathcal{S}_{t-1} \right\}.$$

Under the good event, all of the confidence intervals computed by the policy are valid, and $\mu_i \geq LCB_{i,t}$ for every $t \in [T]$ and $i \in \mathcal{S}_{t-1}$. By Hoeffding inequality, for every fixed $t \in [T]$ and $i \in \mathcal{S}_{t-1}$, we have

$$\mathbb{P}\left( |\mu_i - \widehat{\mu}_{i,t}| > \sqrt{\frac{2\ln\delta^{-1}}{n_{i,t}}} \right) \leq \delta.$$

We use an union bound over $t \in [T]$ and $i \in \mathcal{S}_T$ to get

$$\mathbb{P}\left( \mathcal{E}_T(\delta)^C \right) \leq \delta T^2.$$

Then, we have that

$$\mathbb{E}\left[ R_T(\pi_{LCB}) \right] \leq \mathbb{E}\left[ R_T(\pi_{LCB}) \,|\, \mathcal{E}_T(\delta) \right] + \mathbb{E}\left[ R_T(\pi_{LCB}) \,|\, \mathcal{E}_T(\delta)^C \right] \mathbb{P}\left( \mathcal{E}_T(\delta)^C \right)$$
$$\leq \mathbb{E}\left[ R_T(\pi_{LCB}) \,|\, \mathcal{E}_T(\delta) \right] + \delta T^3,$$

Setting $\delta = \frac{1}{T^3}$, we can now bound the first component of the regret, *i.e.*, the regret under the good event.

We start by proving Equation (3). Action $I_t$ can only be chosen at time $t$ by $\pi_{LCB}$ if $LCB_{I_t,t} \geq LCB_{i_t^\star,t}$. Under the good event, this implies

$$\mu_{i_t^\star} - \mu_{I_t} = \mu_{i_t^\star} \mp LCB_{i_t^\star,t} \mp LCB_{I_t,t} - \mu_{I_t}$$
$$\leq \mu_{i_t^\star} - LCB_{i_t^\star,t}$$
$$\leq \sqrt{\frac{6\ln T^3}{n_{i_t^\star,t}}}.$$

Let $i_t^\star = \mathcal{S}_T^\star(j)$. We sum up the total regret during the $j$-th epoch $\mathcal{I}_j$.

$$\sum_{t \in \mathcal{I}_j} \mu_{\mathcal{S}_T^\star(j)} - \mu_{I_t} = \sum_{\ell=1}^{|\mathcal{I}_j|} \sqrt{\frac{6\ln T^3}{\ell}}$$

$$\leq 2\sqrt{6\ln T^3|\mathcal{I}_j|}.$$

Summing up over the epochs, and noting that $\sum_{j=1}^{\Upsilon^\star}|\mathcal{I}_j| = T$ concludes the proof:

$$\sum_{j=1}^{\Upsilon^\star} 2\sqrt{18\ln T|\mathcal{I}_j|} \leq 2\sqrt{18\Upsilon^\star T\ln T}.$$

To prove Equation (4), we rely on a simple charging scheme. We account the optimal expert at time $t$ to the expert chosen at time $t + T^{2/3}$, called $\widetilde{I}_t$. In the last $T^{2/3}$ rounds, we account for the worst-case cumulative regret of $T^{2/3}$. At time $t + T^{2/3}$, we have $n_{i_t^\star, t+T^{2/3}} \geq T^{2/3}$. Thus,

$$\mu_{i_t^\star} - \mu_{\widetilde{I}_t} = \mu_{i_t^\star} - \mu_{\widetilde{I}_t} \pm LCB_{\widetilde{I}_t, t+T^{2/3}} \pm LCB_{i_t^\star, t+T^{2/3}}$$
$$\leq \mu_{i_t^\star} - LCB_{i_t^\star, t+T^{2/3}}$$
$$\leq \sqrt{\frac{6\ln T}{T^{2/3}}}.$$

Hence, the final regret can be bounded by noting that

$$\sum_{t\in[T]} \left(\mu_{i_t^\star} - \mu_{I_t}\right) \leq \sum_{t\in[T-T^{2/3}]} \left(\mu_{i_t^\star} - \mu_{\widetilde{I}_t}\right) + T^{2/3}$$
$$= T^{2/3}\left(2\sqrt{6\ln T} + 1\right).$$

This concludes the proof. $\qquad\square$

## B. Omitted Proofs for the Adversarial ACE Problem

**Proposition 4.1** (Lower Bound for Adversarial ACE). *Let $\alpha \in (0,1)$. Then, for any policy $\pi$ and any $T \in \mathbb{N}$, there exists an instance in which the expected ordering $\alpha$-regret can be lower bounded as*

$$\mathbb{E}[R_{\alpha,T}(\pi)] \geq \frac{1}{4}\alpha^2 T.$$

*Proof.* Let $M \in [T]$ be set in the following. For simplicity of the analysis, assume that $T$ is a multiple of $M$. Then, let define $t_i = i \cdot M$ for each $i \in [T/M]$ and
$$\mathcal{T} = \{t_i\}_{i\in[T/M]}.$$

We now construct a family of instances. Let $\Phi$ denote the set of all sequences obtained by selecting exactly one expert from each set $\mathcal{S}_{t_i} \setminus \mathcal{S}_{t_{i-1}}$, namely

$$\Phi = \left\{\phi = \{j_i\}_{i\in[T/K]} \in [T]^{T/M} : j_i \in \mathcal{S}_{t_i} \setminus \mathcal{S}_{t_{i-1}}\right\},$$

where we abuse of notation we set $\mathcal{S}_{t_0} = \emptyset$. Given an instance $\phi = \{j_i\}_{i=1}^{T/M} \in \Phi$, we define the corresponding rewards as

$$X_{j,t}^\phi = \begin{cases} 1 & \text{if } t \in \mathcal{T}, j = j_i \text{ and } t_i = t, \\ 0 & \text{otherwise.} \end{cases}$$

By construction two instances $\phi = \{j_i\}_{i\in[T/M]}$ and $\phi' = \{j_i'\}_{i\in[T/M]}$ are identical, and therefore perfectly undistinguishable by any algorithm, up to the first $t_i \in \mathcal{T}$ such that $j_i \neq j_i'$. Hence, for any policy $\pi'$ there exists a policy $\pi$ whose decision at time $t$ are independent from the history up to round $t-1$ that performs at least as good in the worst performing instance in $\Phi$.

It is easy to see that for any such $\pi$, its expected cumulative reward on the worst-case instance satisfies

$$\min_{\phi \in \Phi} \sum_{t=1}^{T} \mathbb{E}[X^{\phi}_{\pi(t),t}] \leq \frac{1}{|\Phi|} \sum_{\phi=\{j_i\} \in \Phi} \sum_{t_i \in \mathcal{T}} \mathbb{P}_{\pi(t_i)}\big(I_{t_i} = j_i\big)$$

$$= \frac{1}{M} \sum_{t_i \in \mathcal{T}} \sum_{j_i \in \mathcal{S}_{t_i} \setminus \mathcal{S}_{t_{i-1}}} \mathbb{P}_{\pi(t_i)}\big(I_{t_i} = j_i\big)$$

$$\leq \frac{1}{M} \sum_{t_i \in \mathcal{T}} 1 = \frac{|\mathcal{T}|}{M} = \frac{T}{M^2},$$

where the probability is with respect to the policy $\pi$.

Now, we show that for any instance $\phi = \{j_i\} \in \Phi$, there is a ranking $\sigma^{\phi}$ that guarantees an high reward. In particular, consider any ordering $\sigma^{\phi}$ such that the top $T/M$ ranked experts are $j_{T/M}, \ldots, j_1$, while the other expert are ordered arbitrarily. Then, the induced policy $\pi_{\sigma^{\phi}}$ on instance $\phi$ achieves cumulative reward

$$\sum_{t=1}^{T} \mathbb{E}[X_{\pi_{\sigma^{\phi}},t}] = |\mathcal{T}| = \frac{T}{M}.$$

Combining the two bounds, we obtain that for at least one instance in $\Phi$ it holds:

$$\sum_{t=1}^{T} \mathbb{E}[X_{\pi(t),t}] \leq \frac{T}{M^2} = \frac{2}{M} \cdot \frac{T}{M} - \frac{T}{M^2}$$

$$= \frac{2}{M} \sum_{t=1}^{T} \mathbb{E}[X_{\pi_{\sigma^{\phi}},t}] - \frac{T}{M^2}.$$

Setting $\alpha = \frac{2}{M}$ concludes the proof. $\qquad\square$

**Theorem 4.2** (Upper Bound for ACE). *For any $\alpha \in [0,1]$, Algorithm 2 guarantees*

$$R_{\alpha,T}(\pi_{\text{adv}}) \leq \mathcal{O}\left(\sqrt{\log(\alpha^{-1})}\alpha^{3/2}T\sqrt{\log(\log(T))}\right).$$

*Proof.* Consider the policy $\pi$ implemented by *Algorithm 2*.

We starting splitting the expected reward of the algorithm into two components: one for long interval and one for short ones. In particular, we observe that:

$$\sum_{t=1}^{T} \mathbb{E}[X_{\pi_{\text{adv}}(t),t}] = \sum_{I \in \mathcal{I}^{\star}} \sum_{t \in I} \mathbb{E}[X_{\pi_{\text{adv}}(t),t}] = \sum_{I \in \mathcal{I}^{\star}_{\text{long}}} \sum_{t \in I} \mathbb{E}[X_{\pi_{\text{adv}}(t),t}] + \sum_{I \in \mathcal{I}^{\star}_{\text{short}}} \sum_{t \in I} \mathbb{E}[X_{\pi_{\text{adv}}(t),t}].$$

**Analysis of Short Intervals**   Consider the term $\sum_{I \in \mathcal{I}^{\star}_{\text{short}}} \sum_{t \in I} \mathbb{E}[X_{\pi_{\text{adv}}(t),t}]$. For all $I \in \mathcal{I}_{\text{short}}$ and for all $t \in I$, by construction $i_t^{\star} \in \mathcal{S}_t \setminus \mathcal{S}_{t-M}$. Hence, at round $t \in I$, with probability $p_1$, Algorithm 2 picks a random expert in $\mathcal{S}_t \setminus \mathcal{S}_{t-M}$. This guarantees that

$$\sum_{I \in \mathcal{I}^{\star}_{\text{short}}} \sum_{t \in I} \mathbb{E}[X_{\pi_{\text{adv}}(t),t}] \geq p_1 \frac{1}{M} \sum_{I \in \mathcal{I}^{\star}_{\text{short}}} \sum_{t \in I} \mathbb{E}[X_{i_t^{\star},t}].$$

**Analysis of Long Intervals**   Consider now an interval $I = \{t_1, \ldots, t_2\} \in \mathcal{I}^{\star}_{\text{long}}$ and recall that $i_{t_1}^{\star}$ is the expert picked by the optimal policy in this interval. We start observing that there exists a regret minimizer $\mathcal{B}_{j_I^{\star}}$ such that $M2^{j_I^{\star}} \leq |I| \leq M2^{j_I^{\star}+1}$. Moreover, this regret minimizer is reset at a round $t'$ such that

$$I \in \{t' - M2^{j_I^{\star}}, \ldots, t' + M2^{j_I^{\star}}\}.$$

This implies that both on the interval $\{t' - M2^{j_I^\star}, \dots, t' - 1\}$ and $\{t', \dots, t' + M2^{j_I^\star}\}$ the expert $i_I^\star$ belongs to the set of experts available to $\mathcal{B}_{j_I^\star}$.

By applying the weakly-adaptive regret guarantees on the two intervals we get:

$$\mathbb{E}\left[\sum_{t \in \{t_1, \dots, t'-1\}} X_{t, i_{t_1}^\star} - r_t(\mathcal{B}_{j_I^\star})\right] \leq \mathcal{O}(\sqrt{|I|\log(|I|)})$$

$$\mathbb{E}\left[\sum_{t \in \{t', \dots, t_2\}} X_{t, i_{t_1}^\star} - r_t(\mathcal{B}_{j_I^\star})\right] \leq \mathcal{O}(\sqrt{|I|\log(|I|)}),$$

where we recall that $r_t(\mathcal{B}_j)$ denotes the rewards of the $j$-th learner at round $t$.

Combining the two results, we get:

$$\mathbb{E}\left[\sum_{t \in I} X_{t, i_{t_1}^\star} - r_t(\mathcal{B}_{j_I^\star})\right] \leq \mathcal{O}(\sqrt{|I|\log(|I|)}).$$

Then, we can split $I$ into $\lceil |I|/M \rceil$ intervals of length at most $K$. On each interval we get that the regret of the meta regret minimizer with respect to following regret minimizer $\mathcal{B}_{j_I^\star}$ is $\mathcal{O}(\sqrt{M\log(M\log(T))})$. Hence, over the whole interval $I$, the regret of the meta learner with respect to learner $j_I^\star$ is at most $\mathcal{O}\left(\frac{|I|}{\sqrt{M}}\sqrt{\log(M\log(T))}\right)$. Then, we get

$$\sum_{I \in \mathcal{I}_{\text{long}}^\star} \sum_{t \in I} \left(\mathbb{E}[r_t(\mathcal{A}_{\text{MASTER}})] - \mathbb{E}[X_{i_t^\star, t}]\right) \leq \sum_{I \in \mathcal{I}_{\text{long}}^\star}\left[\mathcal{O}\left(\frac{|I|}{\sqrt{M}}\sqrt{\log(M\log(T))}\right) + O\left(\sqrt{|I|\log(|I|)}\right)\right]$$

$$\leq \mathcal{O}\left(\frac{T}{\sqrt{M}}\sqrt{\log(M\log(T))}\right) + O\left(\frac{T}{\sqrt{M}}\sqrt{\log(M)}\right)$$

$$\leq \mathcal{O}\left(\frac{T}{\sqrt{M}}\sqrt{\log(M\log(T))}\right),$$

where in the second inequality we used

$$\sum_{I \in \mathcal{I}_{\text{long}}^\star} \mathcal{O}\left(\sqrt{|I|\log(|I|)}\right) = \sum_{I \in \mathcal{I}_{\text{long}}^\star} \mathcal{O}\left(|I|\sqrt{\frac{\log(|I|)}{|I|}}\right) \leq \sum_{I \in \mathcal{I}_{\text{long}}^\star} \mathcal{O}\left(|I|\sqrt{\frac{\log(M)}{M}}\right) \leq \mathcal{O}\left(\frac{T}{\sqrt{M}}\sqrt{\log(M)}\right)$$

**Putting Everything Together**    Now, we are ready to set

$$p_1 = \frac{M}{M+1}, p_2 = \frac{1}{M+1}.$$

Hence, we conclude that Algorithm 2 guarantees expected reward at least

$$p_1 \frac{1}{M} \sum_{I \in \mathcal{I}_{\text{short}}^\star} \sum_{t \in I} \mathbb{E}[X_{i_t^\star, t}] + p_2\left[\sum_{I \in \mathcal{I}_{\text{long}}^\star} \sum_{t \in I} \mathbb{E}[X_{i_t^\star, t}] - \mathcal{O}\left(\frac{T}{\sqrt{M}}\sqrt{\log(M\log(T))}\right)\right]$$

$$\frac{1}{M+1} \sum_{I \in \mathcal{I}_{\text{short}}^\star} \sum_{t \in I} \mathbb{E}[X_{i_t^\star, t}] + \frac{1}{M+1}\left[\sum_{I \in \mathcal{I}_{\text{long}}^\star} \sum_{t \in I} \mathbb{E}[X_{i_t^\star, t}] - \mathcal{O}\left(\frac{T}{\sqrt{M}}\sqrt{\log(M\log(T))}\right)\right]$$

$$\geq \frac{1}{M+1}\left(\sum_{t=1}^{T} \mathbb{E}[X_{i_t^\star, t}]\right) - \mathcal{O}\left(\frac{1}{M+1}\frac{T\sqrt{\log(M\log(T))}}{\sqrt{M}}\right)$$

$$= \frac{1}{M+1} \left( \sum_{t=1}^{T} \mathbb{E}[X_{\pi_{\sigma^\star}(t),t}] \right) - \mathcal{O}\left( \frac{1}{M+1} \frac{T\sqrt{\log(M\log(T))}}{\sqrt{M}} \right),$$

Setting $\alpha = \frac{1}{M+1}$, we get

$$R_{\alpha,T}(\pi) \leq \mathcal{O}\left( \sqrt{\log(\alpha^{-1})}\alpha^{3/2}T\sqrt{\log(\log(T))} \right),$$

concluding the proof. $\square$

## C. Experimental Evaluation

We provide an experimental evaluation of our pessimistic policy $\pi_{LCB}$ for the stochastic ACE problem (Algorithm 1).

We experimentally evaluate our algorithm on instances inspired by the problem of *financial forecasting under regime shifts*. In this problem, the goal is to forecast the evolution of the price of a given stock over time. Specifically, letting $P_t$ denote the price of the stock at day $t$, the objective is to predict the binary price movement

$$Y_t = \mathbb{1}_{\{P_{t+1} > P_t\}},$$

*i.e.*, whether the stock price will increase from day $t$ to day $t+1$.

The forecaster (learner) has access to a pool of experts, where expert $i$ employs an $i$-day look-back rule by predicting

$$\hat{Y}_{i,t} = \mathbb{1}_{\{P_t > P_{t-i}\}},$$

*i.e.*, the expert predicts the direction of the stock price at time $t+1$ by comparing the current price $P_t$ with the price observed $i$ days earlier. The reward of expert $i$ at time $t$ is then defined as

$$X_{i,t} = \mathbb{1}_{\{Y_t = \hat{Y}_{i,t}\}},$$

which indicates whether the expert's prediction is correct. Clearly, this problem can be cast as a stochastic ACE instance. Indeed, expert $i$ deterministically awakens at time $t = i$, since it requires the stock price observed $i$ days earlier in order to make a prediction. Moreover, it is common to model stock prices as evolving stochastically.

### C.1. Real-World Data

We evaluated our algorithm on Yahoo Finance data ($T = 1500$ days, approximately covering the period 2018–2024) (Yahoo Finance, 2024). We compare its performance against the standard UCB algorithm. Performance is measured in terms of cumulative ordering regret and number of switches between experts. We also report the number of swiches performed by an optimal ordering policy as a reference.

Figure 2 reports the results of the experimental evaluation. We observe that the standard UCB algorithm switches experts almost twice every three days, whereas our algorithm successfully tracks market regimes while incurring significantly smaller switching costs. As a result, our algorithm substantially outperforms UCB also in terms of cumulative ordering regret. Notice that, as expected, the UCB algorithm is *not* able to successfully control its cumulative ordering regret, which exhibits a superlinear growth.

### C.2. Simulated Experiments

We also evaluate our algorithm in controlled stochastic environments (20 runs, $T = 5000$) to validate our bounds.

We consider three different kind of instances.

- **First-Best Instance.** In this instance, the first expert to appear is also the best ($\mu_1 = 0.9$), and the others are all equal and slightly worse ($\mu_j$ for all $j \geq 2$). Figure 3 shows our results. An optimistic policy struggles in this type of instances: exploring always results in incurred regret. We simulate the environment 20 times, setting $T = 5000$.

- **Uniform Experts Instance.** In this instance, the expert appearing at time $t$ has its average reward $\mu_t$ sampled from a uniform distribution in $[0.1, 0.9]$. Figure 4 shows our results. An optimistic policy struggles in this type of instances: exploring always results in incurred regret. We simulate the environment 20 times, setting $T = 5000$.

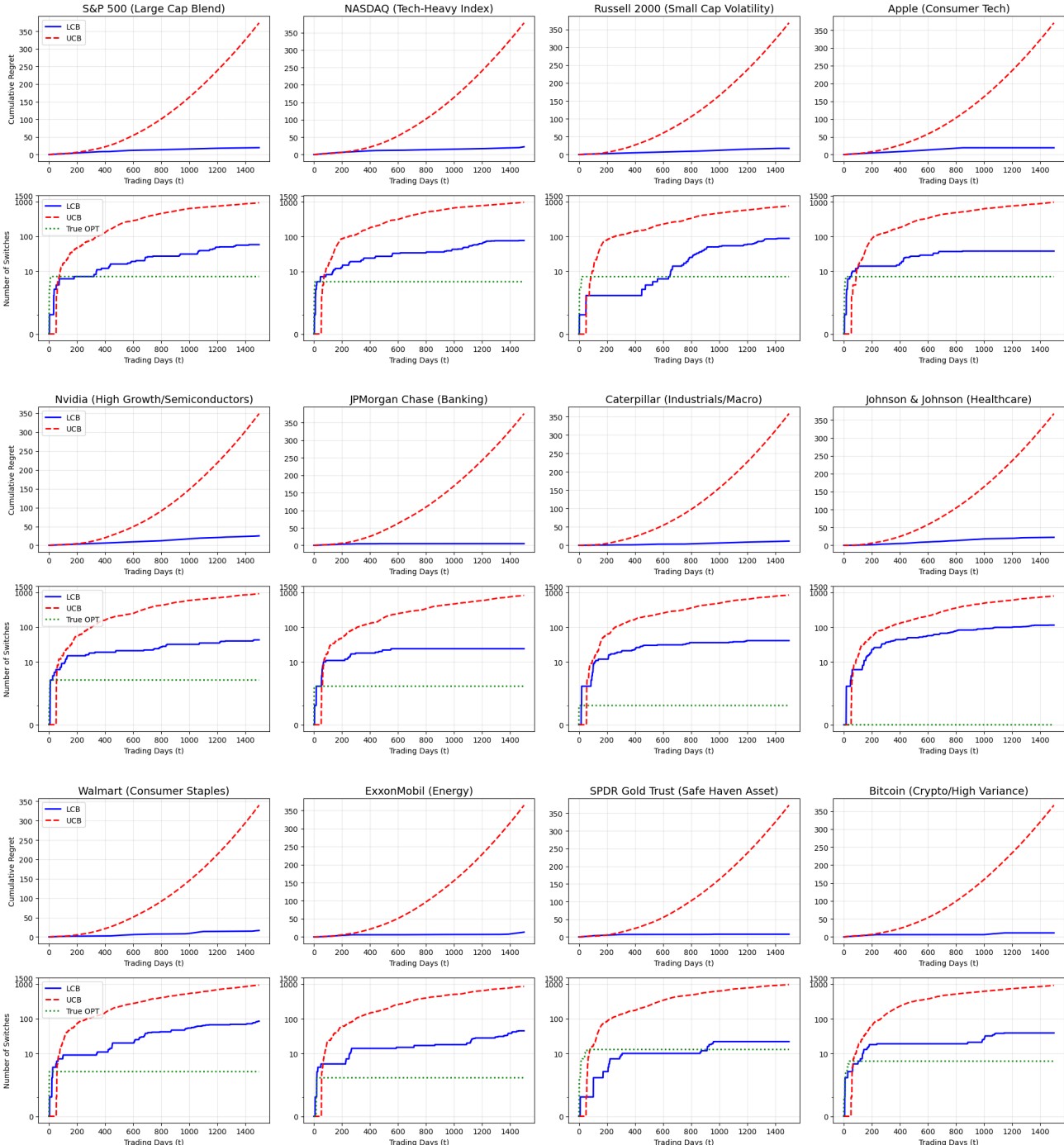

*Figure 2.* Experimental evaluation of $\pi_{LCB}$ (Algorithm 1) against the standard UCB algorithm on real-world data. The plots are organized in pairs, each corresponding to a different stock from the Yahoo Finance dataset. In each pair, the upper plot reports the cumulative regret, while the lower plot reports the number of switches between experts.

- **Frequent Switches Instance.** In this instance, we mimic the lower bound construction by defining a large number of epochs (*e.g.*, $T^{1/5}$ epochs), dividing the trial in epochs and letting the optimum's expected reward increase linearly from 0.1 to 0.9. Figure 5 shows our results. Our algorithm still outperforms UCB.

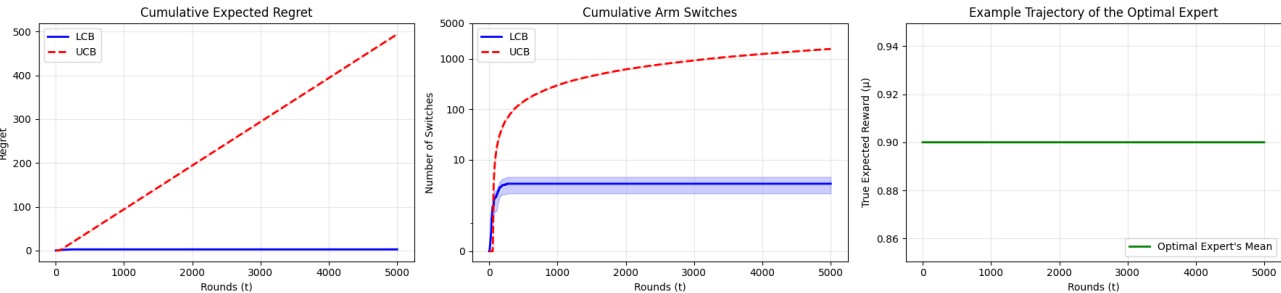

*Figure 3.* Experimental evaluation of $\pi_{LCB}$ (Algorithm 1) against the standard UCB algorithm in controlled stochastic environments: the case of the **first-best instance**.

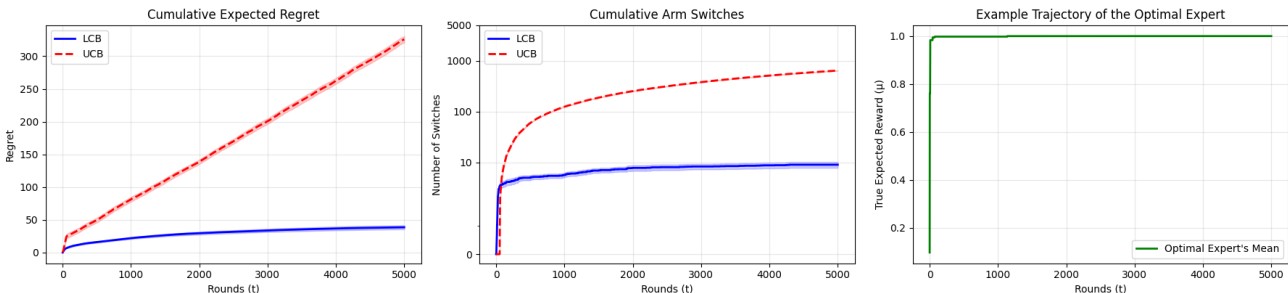

*Figure 4.* Experimental evaluation of $\pi_{LCB}$ (Algorithm 1) against the standard UCB algorithm in controlled stochastic environments: the case of the **uniform experts instance**.

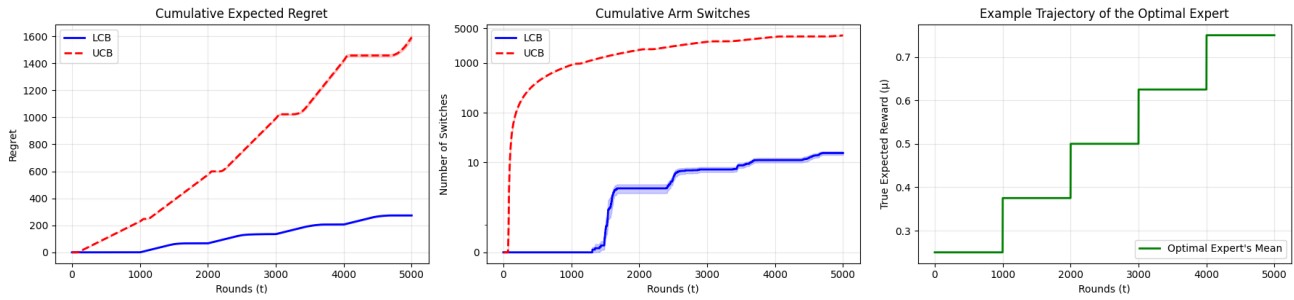

*Figure 5.* Experimental evaluation of $\pi_{LCB}$ (Algorithm 1) against the standard UCB algorithm in controlled stochastic environments: the case of the **frequent switches instance**.

