# OpenReview forum: "Regret Minimization With a Crowd of Awakening Experts"
_ICML.cc/2026/Conference — ICML 2026 regular_

### Official Review · Reviewer_f3cH · 2026-03-05

**Soundness:** 3
**Presentation:** 2
**Significance:** 2
**Originality:** 3
**Overall Recommendation:** 4
**Confidence:** 2

**Summary:**

This paper systematically investigates the Awakening Crowd of Experts (ACE) problem, in which a new expert awakens and joins the set in each round and never leaves thereafter; consequently, the total number of experts can grow up to $K=T$. The authors adopt the ordering policy from the sleeping experts framework as the benchmark and define ordering regret as well as $\alpha$-regret / competitive ratio. The paper studies the ACE problem under both stochastic and adversarial settings. It proves that sublinear regret can still be achieved even when $K=T$. The authors further propose an algorithm based on the Lower Confidence Bound (LCB) principle and provide worst-case upper bounds. In addition, they derive instance-dependent bounds that depend on the number of switches in the optimal ordering policy and establish matching lower bounds. The paper also proves that for any constant $\alpha \in (0,1)$, the $\alpha$-regret must be linear, implying that a trade-off between competitive ratio and additive regret is unavoidable. To address this issue, the authors propose a randomized mixed strategy. Although the work is novel in its formulation and results, the paper still requires further improvement.

**Compliance With Llm Reviewing Policy:**

Affirmed.

**Final Justification:**

The authors' rebuttal has addressed my main concerns regarding the original submission. Therefore, I have updated my assessment and raised my score to a positive rating of 4 (Weak Accept). However, I must note that given my limited familiarity with the specific topic of this research, my confidence is relatively low, so I will not strongly champion the paper for acceptance.

**Key Questions For Authors:**

Q1. It would be beneficial to add experiments to demonstrate the practical performance of the proposed methods, as well as to cover additional scenarios—for example, excessive switching induced by the continual arrival of new experts.

Q2. The paper should also provide a formal analysis (or proof) of the time and space complexity.

Other issues see W1–W3.

**Limitations:**

yes

**Strengths And Weaknesses:**

Strengths：

ST1. Compared with traditional approaches, the monotonically expanding awake set studied by the authors—where experts can only be added and never removed—provides a novel modeling perspective for social proof mechanisms.

ST2. The theoretical analysis is relatively comprehensive, forming a closed theoretical framework that also includes corresponding lower bounds.

ST3.Under the adversarial ACE setting, the paper reveals a structural impossibility result and proposes a corresponding trade-off algorithm. In particular, the linear lower bound on $\alpha$-regret clearly characterizes the inherent trade-off. The proposed algorithm adopts a hybrid structure that combines recent randomized selection with adaptive learning over long intervals, which aligns with the property that the optimal ordering policy can be decomposed into intervals.

Weaknesses：

W1. The paper’s presentation is opaque and difficult to follow, making it hard for readers without substantial background knowledge to understand. Moreover, it does not provide illustrative examples to support the exposition. At a minimum, the paper should include concrete motivating scenarios and explicit problem definitions for both the stochastic and adversarial regimes.

W2. The paper also lacks experiments: it is almost entirely composed of theoretical results and proofs, with no simulations or real-world empirical studies to validate the claims.

W3. In addition, the authors do not provide reproducibility materials, such as accompanying code, to facilitate replication.

---

> ### Author Rebuttal · Authors · 2026-03-30
>
> We thank the Reviewer for their comments and for the effort devoted to reviewing our paper. In what follows, we address the Reviewer's concerns point-by-point.
>
> > Motivating Examples
>
> Section 1.1 details concrete application scenarios like peer feedback systems and social proof marketing.
>
> We introduce an additional motivating example for ACE, which is related to financial forecasting under regime shifts. Given a stock, at day $t$, goal is to predict binary price movement $Y_t = \mathbb{I}(P_{t+1} > P_t)$. Expert $i$ uses an $i$-day lookback rule: $\hat{Y}\_{i,t} = \mathbf{1}(P_t > P_{t-i})$. Requiring $i$ historical prices, expert $i$ deterministically "awakens" at $t=i$. The active set expands as $|\mathcal{S}\_t| = t$. Expert $i$'s reward is $X_{i,t} = \mathbb{I}(\hat{Y}\_{i,t} = Y_t) \in \\{0,1\\} $. As shown in the literature, many markets can be modelled as stochastic.
>
> > Experiments
>
> ### Real-World Data
> We tested our LCB policy on Yahoo Finance data ($T=1500$ days, approx. 2018–2024). Our baseline, standard UCB, switches almost twice every three days, while LCB successfully tracks market regimes with significantly smaller switching costs.
>
> | Asset | $\pi_{LCB}$ Regret | $\pi_{UCB}$ Regret | True OPT Switches ($\Upsilon^*$) | $\pi_{LCB}$ Switches | $\pi_{UCB}$ Switches |
> | :--- | :--- | :--- | :--- | :--- | :--- |
> | **SPX** | 19.60 | 373.85 | 7 | **58** | 937 |
> | **NDX** | 22.39 | 378.23 | 5 | **77** | 990 |
> | **RUT** | 17.44 | 368.23 | 7 | **88** | 761 |
> | **AAPL** | 19.18 | 370.60 | 7 | **38** | 981 |
> | **NVDA** | 25.11 | 349.17 | 3 | **43** | 925 |
> | **JPM** | 4.89 | 376.11 | 2 | **24** | 827 |
> | **CAT** | 11.45 | 358.36 | 1 | **41** | 846 |
> | **JNJ** | 22.36 | 367.86 | 0 | **115** | 792 |
> | **WMT** | 16.65 | 340.04 | 3 | **85** | 936 |
> | **XOM** | 13.17 | 364.34 | 2 | **45** | 855 |
> | **GLD** | 7.63 | 372.80 | 13 | **22** | 968 |
> | **BTC** | 11.18 | 366.49 | 6 | **39** | 920 |
>
> Plots of cumulative regrets and number of switches are at this [[link](https://anonymous.4open.science/r/awakening-crowd-of-experts-plots-6A16/README.md)].
> ### Simulated Data
> We tested the algorithms in controlled stochastic environments (20 runs, $T=5000$) to validate our bounds.
>
> ### 1. First-Best Instance
> In this instance, the first expert to appear is also the best ($\mu_1 = 0.9$), and the others are all equal and slightly worse ($\mu_j = 0.8~~\forall j \ge 2$). An optimistic policy struggles in this type of instance: exploring always results in incurred regret. We simulate the environment $20$ times, setting $T=5000$.
>
> | Metric | $\pi_{LCB}$ | $\pi_{UCB}$ (Baseline) |
> | :--- | :--- | :--- |
> | **Final Cumulative Regret** | 2.59 ± 1.11 | 494.12 ± 0.11 |
> | **Total Arm Switches** | 3.35 ± 1.20 | 1553.15 ± 6.87 |
>
> ### 2. Uniform Experts
> Here, the expert appearing at time $t$ has its average reward $\mu_t$ sampled from a uniform in $[0.1,0.9]$. An optimistic policy struggles in this type of instance: exploring always results in incurred regret. We simulate the environment $20$ times, setting $T=5000$.
>
> | Metric | $\pi_{LCB}$ | $\pi_{UCB}$ (Baseline) |
> | :--- | :--- | :--- |
> | **Cumulative Regret** | 38.50 ± 3.04 | 325.89 ± 4.85 |
> | **Arm Switches** | 8.95 ± 1.18 | 640.70 ± 4.41 |
>
> ### 3. Frequent Switches Instance
> In this instance, we mimic the lower bound construction by defining a large number of epochs (e.g., $T^{1/5}$), dividing the trial into epochs, and letting the optimum's expected reward increase linearly from $0.1$ to $0.9$. LCB still outperforms UCB.
>
> | Metric | $\pi_{LCB}$ (Proposed) | $\pi_{UCB}$ (Baseline) |
> | :--- | :--- | :--- |
> | **Cumulative Regret** | 273.16 ± 3.89 | 1591.44 ± 15.60 |
> | **Arm Switches** | 15.40 ± 1.28 | 3589.25 ± 23.05 |
>
> Plots of cumulative regrets and number of switches are at this [[link](https://anonymous.4open.science/r/awakening-crowd-of-experts-plots-6A16/README.md)].
>
> **We will complete the submission, including the complete codebase used to conduct this experimental campaign, together with parameters and seed for reproducibility**
>
> > Time and space complexity.
>
> **Stochastic case:** Per-round time and space complexity of LCB are $O(T)$. At each round, the algorithm performs a constant number of operations and stores one value (empirical mean) per expert.
>
> **Adversarial case:** The per-round time complexity of the adversarial algorithm can be computed as the sum of the per-round time complexity of the regret minimizers used. In the worst case (the last round), there are $\sum_{j=0}^{\log_2(T/M)} M 2^j = O(T)$ update operations for the base regret minimizers and $O(\log_2(T/M))$ operations for the master regret minimizer. The space complexity can be computed analogously.

---

> > ### Author Rebuttal · Reviewer_f3cH · 2026-04-03
> >
> > Thank you for the rebuttal. It has largely addressed my concerns, I have decided to raise my score to 4.

---

### Official Review · Reviewer_sbNd · 2026-03-05

**Soundness:** 3
**Presentation:** 4
**Significance:** 3
**Originality:** 2
**Overall Recommendation:** 5
**Confidence:** 3

**Summary:**

The sleeping expert formulation introduced in [1] considers the problem of learning with experts that can be either awake or asleep (not available). However, if the number of experts is of order $T$ (the horizon), the upper regret bounds become vacuous for the algorithms studied in the literature. Therefore, this paper considers "awakening experts", a special case of the sleeping experts problem, where a new expert awakens at every round. This simplification of the problem allows to obtain sublinear regret bounds in the stochastic setting. The case where experts are adversarial is harder and the paper proves that for any constant $\alpha \in (0,1)$, the regret is at least of order $\alpha^2 T$. In that case, it is also proven that a competitive ratio of order $log log T$ can be obtained.

[1] Kleinberg, R., Niculescu-Mizil, A., & Sharma, Y. (2010). Regret bounds for sleeping experts and bandits. Machine learning, 80(2), 245-272.

**Compliance With Llm Reviewing Policy:**

Affirmed.

**Final Justification:**

The authors' rebuttal solved my concerns.

**Key Questions For Authors:**

Questions:
- Could the results be adapted with slightly different "awakening" structures: typically, could it be possible to handle a case where the learner starts with $T$ experts and one of them "falls asleep" at each round?
- Are there cases that bridge the gap between the general sleeping experts problem with linear upper regret bounds and this stronger result for the specific/structured case of awakening experts?

**Limitations:**

Yes.

**Strengths And Weaknesses:**

Strengths:
- I find the paper extremely well written: the problem of awakening experts is nicely motivated with User Generated Content platforms after a clear introduction of the sleeping experts problem. Everything is nice to read and easy to follow.
- The statements of the results are rigorous and clear.
- I think that it tackles an interesting problem in the scope of learning with experts.

Weaknesses:
- Even though the problem is interesting, the scope and the originality of the paper remain a bit limited (which should not be a reason for non acceptance to me).
- The studied problem has a strong structure, and it could be interesting to explore the question of how it could be relaxed.

---

> ### Author Rebuttal · Authors · 2026-03-31
>
> We thank the Reviewer for their comments and for the effort devoted to reviewing our paper.
>
> > Even though the problem is interesting, the scope and the originality of the paper remain a bit limited (which should not be a reason for non-acceptance to me).
>
> We believe that the ACE problem is indeed a natural variation of sleeping bandits that captures the structure of several real-world problems and has been neglected so far by the literature. Moreover, we believe that the techniques developed to tackle regret minimization in the ACE problem, in particular those in the adversarial setting, are original and considerably depart from those commonly used in the sleeping bandits literature.
>
> > Could the results be adapted with slightly different "awakening" structures: typically, could it be possible to handle a case where the learner starts with experts and one of them "falls asleep" at each round?
>
> We believe that the Reviewer might be referring to the "Dying Experts" setting [1], that we mentioned in Section 1.3 of the paper. This setting is, in a certain sense, the opposite of ours. There are some crucial differences in the two settings, indeed, as it can be seen in [1], the employed techniques and the resulting bounds are quite different.
>
> > The studied problem has a strong structure, and it could be interesting to explore the question of how it could be relaxed. / Are there cases that bridge the gap between the general sleeping experts problem with linear upper regret bounds and this stronger result for the specific/structured case of awakening experts?
>
> We agree with the Reviewer that it could be interesting to study ACE problems with a relaxed structure, so as to capture even more real-world settings. One possibility could be the study of ACE problems in which more than one expert may awaken at each round, while another possibility could be addressing settings in which there are some rounds where no expert awakens. We also refer the Reviewer to the response to Reviewer ZRg9 for additional details on how to adapt our results in the stochastic setting to these cases. We believe that both directions are interesting for future works.
>
> [1] Shayestehmanesh, H., Azami, S., & Mehta, N. A. (2019). Dying experts: Efficient algorithms with optimal regret bounds. Advances in Neural Information Processing Systems, 32.

---

> > ### Author Rebuttal · Reviewer_sbNd · 2026-04-02
> >
> > I thank the authors for their answer which solved my concerns. I increased my score accordingly.

---

### Official Review · Reviewer_zdDi · 2026-03-12

**Soundness:** 3
**Presentation:** 3
**Significance:** 3
**Originality:** 3
**Overall Recommendation:** 4
**Confidence:** 3

**Summary:**

The paper introduces the Awakening Crowd of Experts (ACE) problem, an online learning scenario where the number of experts grows linearly with time ($K=T$).
Key Highlights
1 Addressing the "Crowd" Challenge: Standard algorithms yield vacuous (linear) regret when the expert pool is too large. By utilizing the "awakening" property (experts stay once they appear), the authors achieve sublinear regret.
2 Stochastic Setting: The authors propose the $\pi_{LCB}$ algorithm. Moving away from traditional "optimistic" approaches, it uses "unusual pessimism" (Lower Confidence Bounds) to ensure a new expert is only chosen after it proves to be better than established ones.
3 Adversarial Setting: Since standard regret is impossible to minimize here, the paper optimizes $\alpha$-regret and achieves a competitive ratio of $\tilde{\mathcal{O}}(\log \log T)$ using a meta-learning framework.
4 Practical Value: This models real-world platforms like Reddit or social proof marketing, where the system must identify high-quality newcomers without being distracted by an overwhelming "crowd" of unproven options.

**Compliance With Llm Reviewing Policy:**

Affirmed.

**Key Questions For Authors:**

Please refer to the "Weaknesses"

**Limitations:**

Please refer to the "Weaknesses"

**Strengths And Weaknesses:**

Strengths
1 The paper provides solid theoretical analysis, including both upper and lower bounds for the proposed problem, which offers a relatively complete theoretical characterization of the setting.
2 The manuscript is generally well written and clearly structured, making the main ideas and technical results relatively easy to follow.

Weaknesses
1 Conservative switching may limit exploration of new experts.
The pessimistic switching strategy delays adopting new experts to avoid overestimation. In practical systems, however, such conservatism may reduce the exposure of newly arrived experts and potentially affect their incentives or participation. It would be useful to discuss whether additional exploration mechanisms could alleviate this issue.
2 Strong assumptions on expert dynamics.
The model assumes that experts only arrive and never leave, and that exactly one new expert appears in each round (i.e., the number of experts grows linearly with time). These assumptions are central to the analysis but may be restrictive in real-world systems where arrivals can be irregular and experts may become inactive.
3 Stationary expert ability assumption.
The framework assumes that each expert’s underlying ability is fixed over time. In many practical scenarios, expert performance may evolve due to learning, feedback, or environmental changes. It remains unclear how the proposed approach would behave under such non-stationary conditions.

---

> ### Author Rebuttal · Authors · 2026-03-31
>
> We thank the Reviewer for their comments and for the effort devoted to reviewing our paper.
>
> > 1. Conservative switching may limit exploration of new experts. The pessimistic switching strategy delays adopting new experts to avoid overestimation. In practical systems, however, such conservatism may reduce the exposure of newly arrived experts and potentially affect their incentives or participation. It would be useful to discuss whether additional exploration mechanisms could alleviate this issue.
>
> It can be proven that a standard optimistic algorithm incurs linear regret in the ACE setting. If we consider a standard UCB algorithm (with the slight modification that expert $t$ cannot be chosen at time $t$, otherwise the policy would always try the new one), we can construct an instance in which the expected regret is linear in the following way. Let the first expert be optimal through all $T$ rounds with mean $\mu_1 = \frac{1}{2}+\Delta$, for some $\Delta>0$. All subsequent experts have means $\mu_j = 0.5$ for $j \ge 2$. At time $t$, the expert arrived in $t-1$ generated a reward $X_{t-1,t}=1$ with probability $\frac{1}{2}$. At this point, the UCB policy surely selects expert $t-1$ at time $t+1$, incurring an expected regret of $\Delta$. To sum up, at each round, with constant probability, we have a regret of $\Delta$, which rolls up to a linear regret over $T$ rounds.
>
> This can also be observed from an empirical perspective, as reported in the answer to Reviewer f3cH. LCB always outperforms UCB, which incurs linear regret in almost every instance.
>
> We would like to point out that being conservative is a key characteristic for every algorithm suited for the ACE setting. Unlike standard bandits, the price of exploration here is higher: not because less information can be gained, but because the decision space expands at the same speed. Indeed, **LCB is provably optimal for this setting.**
>
> > 2. Strong assumptions on expert dynamics. The model assumes that experts only arrive and never leave, and that exactly one new expert appears in each round (i.e., the number of experts grows linearly with time). These assumptions are central to the analysis but may be restrictive in real-world systems where arrivals can be irregular and experts may become inactive.
>
> We agree with the Reviewer that it could be interesting to study ACE problems with a relaxed structure, so as to capture even more real-world settings. One possibility could be the study of ACE problems in which more than one expert may awaken at each round, while another possibility could be addressing settings in which there are some rounds where no expert awakens. We also refer the Reviewer to the response to Reviewer ZRg9 for additional details on how to adapt our results in the stochastic setting to the case in which multiple experts may awaken at each round. We believe that both directions are interesting for future works. Finally, notice that if one allows experts to awake and then fall asleep again, one falls back to the sleeping experts setting, where linear regret is unavoidable if the total number of experts grows linearly in the time horizon.
>
> > 3. Stationary expert ability assumption. The framework assumes that each expert’s underlying ability is fixed over time. In many practical scenarios, expert performance may evolve due to learning, feedback, or environmental changes. It remains unclear how the proposed approach would behave under such non-stationary conditions.
>
> Notice that, in the adversarial setting, each expert's ability may vary arbitrarily across rounds. Of course, one may imagine a setting in which experts' abilities may vary across rounds in a way that is not fully adversarial, but rather closer to the stochastic setting (e.g., distributional shifts). We believe that this is an interesting direction for future work, but it is out of the scope of a first paper on the ACE setting.

---

### Official Review · Reviewer_ZRg9 · 2026-03-12

**Soundness:** 3
**Presentation:** 3
**Significance:** 2
**Originality:** 3
**Overall Recommendation:** 5
**Confidence:** 4

**Summary:**

The paper introduces the Awakening Crowd of Experts (ACE) problem, where a new expert arrives at each round and never leaves. This is a special case of the sleeping experts problem for K = T experts. Existing sleeping experts bounds mostly hold when K is constant with respect to T, so they are vacuous in this case. The authors use ordering regret as a benchmark (as is standard in the sleeping experts literature) and study both the stochastic and the adversarial setting. In the stochastic setting they provide matching lower and upper bounds. In the adversarial setting, as a lower bound, they show that sublinear $\alpha$-regret is impossible for constant $\alpha$ and they provide an algorithm that, when applied in the regime OPT = $\Theta(T)$ gives an $O(\log \log T)$ competitive ratio.

**Compliance With Llm Reviewing Policy:**

Affirmed.

**Final Justification:**

This paper makes an interesting contribution on a new experts setting and the rebuttal has addressed my concerns. Therefore I have increased my score to 5 (Accept).

**Key Questions For Authors:**

* (Q1) I think it's worth discussing a bit more why we should care about having a seemingly super-linear regret upper bound. It was not clear from the beginning and it is worth mentioning the connection to bounding the competitive ratio earlier on

* (Q2) Do the results for the adversarial case extend to multiple expert arrivals as well? Also, do the results (both stochastic and adversarial) extend trivially to the case where there might not be an awakening expert at every time t ? (This would still provide an extension over the standard sleeping experts problem to the regime where the awakening experts is large but not $\Theta(T)$.)

**Limitations:**

yes

**Strengths And Weaknesses:**

# Strengths

* The paper is well-written and the problem well-motivated and natural.

* Full study with matching upper and lower bounds for the stochastic setting and almost-matching bounds for the adversarial setting.

* The algorithm for the stochastic setting is elegant and simple, and it is a nice example of a case where we need to invert the usual UCB principle. It is The algorithm for the adversarial setting is also quite elegant.

# Weaknesses

* (W1) I think it is worth adding a more extensive discussion of why the ordering regret baseline makes sense in the awakening experts problem, and whether there are other natural regret benchmarks that could be used.

* (W2) It would be informative to add some more discussion of prior works on adaptive regret and hierarchical regret minimizer algorithms. For example, [1] appears to be directly related. Is there any connection with the goals/techniques of [2] ? Is there any connection to the TreeSwap algorithm [3] ?

* (W3) The notation in the proof (both the sketch and full appendix proof) of Theorem 3.2 has some mistakes. For example, shouldn't the inequality bounding the absolute value of $\mu_i$ and $\hat{\mu}_{i,t}$ be in the opposite direction? Also the way you bound the per-iteration regret after this doesn't make sense. There seem to be some mixed subscripts etc.

* (W4) (minor) Proposition 2.1 is trivial and does not need to have its proof in the main paper body.

# References

[1] Strongly Adaptive Online Learning.
Amit Daniely, Alon Gonen, Shai Shalev-Shwartz. Proceedings of the 32nd International Conference on Machine Learning, 2015.

[2] Dynamic Regret Bounds for Online Omniprediction with Long Term Constraints.
Yahav Bechavod, Jiuyao Lu, Aaron Roth, arxiv, 2025

[3] From External to Swap Regret 2.0: An Efficient Reduction for Large Action Spaces. Yuval Dagan, Constantinos Daskalakis, Maxwell Fishelson, and Noah Golowich. In Proceedings of the 56th Annual ACM Symposium on Theory of Computing (STOC 2024).

---

> ### Author Rebuttal · Authors · 2026-03-31
>
> We thank the Reviewer for their comments and for the effort devoted to reviewing our paper.
>
> > (W1)
>
> We thank the Reviewer for this suggestion and will include a brief discussion on the various notions of regret in the final manuscript.
>
> In our specific setting, the standard definitions of external regret, policy regret, and ordering regret (as defined in [1]) largely collapse due to the nested structure of the available expert sets. In our model, the active subsets of experts are restricted to the collection $\{S_t\}_{t=1}^T$ where $S_t=\{1,\ldots,t\}$.
>
> **Policy regret vs. ordering regret**
> Given a policy in the space $\pi$ in $\Pi= \{\{1,\ldots,t\}\mapsto i^*_t\in [t], t\in [T]\}$, we can define a ordering $\sigma$, such that the expert choosen by the policy $\pi$ is equal to the expert choosen by the ordering $\sigma$ for all active experts subset.
>
> **External regret vs. ordering regret**
> The notion of external regret in our case becomes $R^{ext}\_T(k)= \sum_{t=1}^T(\ell_t(k_t)-\ell_t(k))\mathbb{I}(k\in S_t)= \sum_{t=k}^T(\ell_t(k_t)-\ell_t(k))$.
> By definition, the optimal ordering $\sigma^\*$ minimizes the loss over all sequences of active sets. If there existed an expert $k$ such that the external regret was lower than the ordering regret, we could construct a new ordering $\sigma'$ that prioritizes $k$, contradicting the optimality of $\sigma^*$.
> Consequently, in our setting, a bound on the ordering regret effectively provides a bound on the external regret
>
> [1] Gaillard, Pierre, Aadirupa Saha, and Soham Dan. "One arrow, two kills: A unified framework for achieving optimal regret guarantees in sleeping bandits." International Conference on Artificial Intelligence and Statistics. PMLR, 2023.
>
> > (W2)
>
> We appreciate the Reviewer's suggestion and will expand the literature comparison in the final version of the paper, specifically addressing the suggested references.
>
> Regarding [1], we acknowledge its relevance to our setting; their techniques for enforcing strongly adaptive budget balance on a regret minimizer could, in principle, be adapted to replace our Algorithm 3 and its subroutines. However, employing [1] would generate an unavoidable $\log(T)$ multiplicative factor in the cumulative regret, which is worse than the $\log\log(T)$ we were able to achieve.
>
> [2,3] share the definition of dynamic regret with our adversarial setting. However, they deal with significantly different problems, and the techniques cannot be applied to our setting as we want to bound the $\alpha$-regret in a very specific setting with a large action space.
>
> > (W3)
>
> We thank the Reviewer for having spotted the reversed inequality, which we have already fixed. It is not clear to us which point of the proof, in particular, does not make sense for the Reviewer. We kindly ask the Reviewer to indicate it, so we can discuss it/fix it in case of confounding notation.
>
> > (Q1)
>
> Having super-linear regret upper bound is somehow limiting when the optimum is linear. In fact, when the optimum is sufficiently large, the algorithm can aim for a constant competitive ratio. Specifically, Corollary 4.3, with regret linear, would guarantee competive ration $\Omega(1)$, while with super-linear regret, in those same instances, we can guarantee only $\Omega(1/\log(\log(T)))$ competive ratio, which is decreasing in $T$.
>
> We will be sure to expand the discussion on the relevance of the super linear regret in the introuduction, to improve the genaral flow of the presentation.
>
> > (Q2)
>
> In the stochastic setting, this scenario is already covered:
> * The lower bound forces two experts to awake every $T^{2/3}$ rounds (totaling $\mathcal{O}(T^{1/3})$ experts); no additional experts are strictly needed.
> * Upper bounds maintain the same order regardless of expert count. If $g(T) > T$, experts awake, it is only paid inside the logarithmic term (for the union bound). If $g(T) < T$, results hold. Since $g(T) \ge \Upsilon^*$, a constant expert count (e.g., $g(T) = K$) retrieves the $\mathcal{O}(\sqrt{TK})$ bound.
>
> In the adversarial case, guarantees rely on the fact that even short-lived optima (window $< K$) are selected with non-negligible probability by uniformly sampling from the $K$ most recent arrivals. As the arrival rate increases, the chance of selecting the "best" arm from this expanding pool decreases, degrading the competitive ratio linearly. This assumes a uniform arrival rate. As in the stochastic case, arrival distribution matters: if all extra experts arrive at $t=1$, full feedback negates their impact. Conversely, an arrival rate below one per round could improve performance, as fewer recent experts increase the probability of picking a short-lived optimum.

---

> > ### Author Rebuttal · Reviewer_ZRg9 · 2026-04-03
> >
> > Thank you for all the clarifications!
> >
> > Regarding the proof, I was referring to second column in line 227 where you write
> >
> > $$
> > \mu_{i_t^\star} - \mu_{I_t} = \mu_{i_t^\star} - \mu_{I_t} \pm \dots
> > $$
> >
> > I don't understand what you mean to say there.
> >
> > Also what is $\tilde{I}_t$ ? Have you defined it somewhere?

---

> > > ### Author Response · Authors · 2026-04-03
> > >
> > > $\tilde{I}\_t$ is defined at line 271, left column. It corresponds to the expert chosen by the policy $\pi_{LCB}$ at time $t+T^{2/3}$. It is convenient to define it since the proof makes use of a charging argument, where we compare the optimum chosen at time $t$ with the expert chosen by $\pi_{LCB}$ at time $t+T^{2/3}$ (i.e., $\tilde{I}\_t$) instead of the expert chosen at time $t$ (i.e., $I_t$). Note that the total cumulative regret can be bounded by doing this for the first $T-T^{2/3}$ rounds, then we just pay $T^{2/3}$ additional regret (line 235, right column).
> > >
> > > The step at line 227 is a standard mathematical trick where we equal a quantity to the same quantity plus a term minus the same term. This is standard in bandit theory, as it is a way to include lower (or upper) confidence bounds in your analysis and use the fact that the expert selected by your algorithm is the one having the largest, dominating the bound of the optimum. In particular, in our case:
> > > 1. We want to bound the difference between the optimal mean at time $t$ and the mean of the expert we selected at time $t+T^{2/3}$, namely $\mu_{i_t^\star}-\mu_{\widetilde{I}_t}$. If we can do this, we can just sum over $t \in [T-T^{2/3}] $, add $T^{2/3}$, and bound our total regret.
> > > 2. If $\pi\_{LCB}$ selects $\widetilde{I}_t$ at time $t+T\^{2/3}$, that means that its LCB is larger than the LCB of $i\_t\^\*$ at the same time. Note that $i\_t\^*$ is of course awake at time $t+T\^{2/3}$, otherwise it couldn't have been optimal at time $t$.
> > > 3. Moreover, at time $t+T^{2/3}$ the expert $i_t^*$ has already been observed for at least $T^{2/3}$ samples.
> > > 4. When bounding the instantaneous regret in stochastic online learning, we usually want to bound the difference between the means of two actions (e.g., the optimal one and the one chosen by the policy) with the width of the confidence interval. To do so we rely on the trick of line 227, which can also be written as:
> > > $$ \mu_{i\_t\^\star}-\mu\_{\widetilde{I}\_t} = \mu_{i\_t\^\star}-\mu\_{\widetilde{I}\_t} + \underbrace{LCB_{\widetilde{I}\_t,t+T\^{\frac{2}{3}}}-LCB_{\widetilde{I}\_t,t+T\^{\frac{2}{3}}}}\_{=0} + \underbrace{LCB_{i\_t^\star,t+T\^{\frac{2}{3}}} - LCB_{i\_t^\star,t+T\^{\frac{2}{3}}}}\_{=0} $$
> > > Then, we just permutate the terms and obtain:
> > > $$ = \mu_{i\_t\^\star}-LCB_{i\_t^\star,t+T\^{\frac{2}{3}}} + \underbrace{LCB_{\widetilde{I}\_t,t+T\^{\frac{2}{3}}} - \mu\_{\widetilde{I}\_t}}\_{\le 0 \text{ by the definition of Lower Confidence Bound}} + \underbrace{LCB_{i\_t^\star,t+T\^{\frac{2}{3}}} - LCB_{\widetilde{I}\_t,t+T\^{\frac{2}{3}}}}\_{\le 0 \text{ by the definition of the $\pi\_{LCB}$ policy (see also point 2.)}} $$
> > > and the instantaneous regret can be finally bounded as
> > > $$ \le \mu_{i\_t\^\star}-LCB\_{i\_t^\star,t+T\^{\frac{2}{3}}} \le \sqrt{6\ln T/T\^{2/3}}$$
> > > by the definition of LCB and the fact that $i\_t\^\star$ already generated at least $T^{2/3}$ samples, as stated in point 3.
> > >
> > > The proof is then concluded by summing over $t$ and excluding the last $T^{2/3}$ rounds, as described in the first paragraph of this rebuttal.
> > >
> > > We hope this answer completely resolved your concerns! Thanks again for your time and effort in reviewing our paper.

---

### Decision · Program_Chairs · 2026-04-30

**Decision:**

Accept (regular)

**Comment:**

This paper introduces the Awakening Crowd of Experts (ACE) problem, where a new expert joins each round and never leaves, resulting in up to T experts. The authors provide matching upper and lower bounds in the stochastic setting and an O(log log T) competitive ratio in the adversarial setting where sublinear regret is impossible. All four reviewers found the problem well-motivated and the theoretical contributions solid. Concerns raised were minor and largely presentational and reviewers confirmed they were resolved by author's rebuttal.